

# A non-unitary bulk-boundary correspondence: Non-unitary Haagerup RCFTs from S-fold SCFTs

Dongmin Gang[1⋆], Dongyeob Kim[1,2†] and Sungjay Lee[3‡]

**1** Department of Physics and Astronomy & Center for Theoretical Physics,
Seoul National University, 1 Gwanak-ro, Seoul 08826, Korea
**2** Department of Physics, Princeton University, Princeton, NJ 08544, USA
**3** Korea Institute for Advanced Study, 85 Hoegiro,
Dongdaemun-Gu, Seoul 02455, Korea

⋆ arima275@snu.ac.kr , † dk2846@princeton.edu , ‡ sjlee@kias.re.kr

## Abstract

We introduce a novel class of two-dimensional non-unitary rational conformal field theories (RCFTs) whose modular data are identical to the generalized Haagerup-Izumi modular data. Via the bulk-boundary correspondence, they are related to the three-dimensional non-unitary Haagerup topological field theories, recently constructed by a topological twisting of three-dimensional $\mathcal{N} = 4$ rank-zero superconformal field theories (SCFTs), called S-fold SCFTs. We propose that, up to the overall factors, the half-indices of the rank-zero SCFTs give the explicit Nahm representation of four conformal characters of the RCFTs including the vacuum character. Using the theory of Bantay-Gannon, we can successfully complete them into the full admissible conformal characters of the RCFTs.

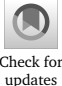

## 1   Introduction

A characteristic property of (2+1)-dimensional topological matters is the presence of chiral gapless states on their boundary. It is well-studied that, when a topological phase in the bulk can be described by a unitary topological quantum field theory (TQFT), the universal behaviors of gapless boundary states can be captured by a two-dimensional unitary rational conformal field theory (RCFT). This connection is often referred to as the bulk-boundary correspondenc [1, 2]. A prominent example is the correspondence between the three-dimensional Chern-Simons theories and the two-dimensional Wess-Zumino-Witten (WZW) models. The Chern-Simons theories are low-energy effective theories for certain quantum Hall systems, typical many-body systems with a mass gap.

The bulk-boundary correspondence however becomes less clear in non-unitary cases, primarily due to the limited understanding of the physical nature of the non-unitary TQFTs. Mathematically, the non-unitary TQFTs can be defined in a manner nearly identical to the unitary ones with certain unitarity conditions being relaxed. In particular, the bulk observables in both theories are encoded in rigid mathematical structure known as the modular tensor category (MTC) [3, 4]. However, the manifestation of a non-unitary TQFT equipped with MTC in a physical system has been a formidable challenge until the recent proposals [5, 6].

Specifically, it was argued that a family of non-unitary TQFTs can be realized as a topological twisting of an exotic class of three-dimensional $\mathcal{N} = 4$ superconformal field theories (SCFTs), called the rank-zero theories. Those rank-zero theories can be characterized by the absence of both Coulomb and Higgs branches, and have an isolated vacuum. Despite the extensive studies of the landscape of the three-dimensional $\mathcal{N} = 4$ SCFTs, this class of exotic SCFTs has largely remained unexplored. This is essentially because most of them are not connected to weakly-coupled UV theories with manifest $\mathcal{N} = 4$ supersymmetry by RG flows. Rather, their $\mathcal{N} = 4$ supersymmetry is accidental in the infrared limit, namely their SUSY get enhanced dynamically.

In three dimensions, the $\mathcal{N} = 4$ SUSY is the minimal SUSY that admits a topological twisting. When a given theory is topologically twisted, the spin-statistics theorem is no longer valid and the unitarity becomes violated. For the theories of rank zero, the topological twisting results in genuine (semi-simple) TQFTs associated with MTC. Thus, one can expect that the topologically twisted $\mathcal{N} = 4$ rank-zero theories physically realizes the non-unitary TQFTs. We summarize the chains of correspondences in the following diagram:

$$\boxed{\text{Rank-zero SCFT}} \xrightarrow{\text{top'l twisting}} \boxed{\text{Non-unitary TQFT}} \xrightarrow{\text{bulk-boundary}} \boxed{\text{Non-unitary RCFT}}$$

In the present work, we examine the non-unitary version of bulk-boundary correspondence for the S-fold SCFTs, defined in (1), as concrete examples. As demonstrated in [7], the topological twisting of such S-fold SCFTs gives rise to the TQFTs with modular data generalizing the non-unitary Haagerup-Izumi modular data. For this reason, we refer to them as the generalized Haagerup TQFTs. Is there a boundary RCFT corresponding to a generalized Haagerup TQFT? To see this, we utilize the so-called Riemann-Hilbert method [8, 9] to explicitly construct the conformal characters compatible with a given generalized Haagerup-Izumi modular data. Herein, the half-index computation of the aforementioned rank-zero theories play a key role in making this approach effective. Note that the central charge read off from the vacuum

conformal character is always negative, which is consistent with the fact that the modular data of our interest give negative quantum dimensions. It strongly supports that there exist non-unitary RCFTs associated with the Haagerup TQFTs, and suggests that the non-unitary version of the bulk-boundary correspondence actually does work.

The rest of the paper is organized as follows. In section 2, we introduce the S-fold SCFTs labeled by an integer $k \geq 3$ and its topologically twisted theories. We also propose a dual field theory description of the SCFTs in terms of Abelian gauge fields. We review the computation of modular data of the topologically twisted theories and explain why we call them generalized non-unitary Haagerup TQFTs. In section 3, we introduce generalized non-unitary Haagerup RCFTs, which live on the boundary of the TQFTs. From the computation of half-indices using the Abelian dual description, we propose fermionic sum representations of some characters of the non-unitary RCFTs. We then verify that the characters can be completed into full admissible RCFT characters by using either modular linear differential equation (MLDE) or Riemann-Hilbert method. We present the full characters up to $k = 11$ but the analysis can be in principle extended to arbitrarily higher $k$. In appendix, we give a derivation of the dual Abelian description using 3D-3D correspondence.

## 2 3D S-fold SCFTs and generalized non-unitary Haagerup TQFTs

### 2.1 S-fold SCFT $\mathcal{S}_k$

Here we introduce two UV descriptions of the S-fold SCFTs. One is using $T[SU(2)]$ theory which is the conventional one. The other is based on $\mathcal{N} = 2$ Abelian gauge fields coupled to chiral multiplets. The Abelian description will be used in defining the generalized Haagerup RCFTs, the main hero of this paper, and computing the half-indices, which give characters of the RCFTs .

#### 2.1.1 From $T[SU(2)]$ theory

We define the S-fold superconformal field theory (SCFT) $\mathcal{S}_k$ by

$$
\begin{aligned}
\mathcal{S}_k &:= \frac{T[SU(2)]}{SU(2)_k^{\mathrm{diag}}} \\
&:= (\text{Gauging diagonal } SU(2) \text{ of } T[SU(2)] \text{ theory} \\
&\qquad \text{with } \mathcal{N} = 3 \text{ Chern-Simons term of level } k).
\end{aligned}
\tag{1}
$$

The $T[SU(2)]$ theory is a 3D $\mathcal{N} = 4$ SQED with two hypermultiplets [10]. The theory has manifest $U(1) \times SU(2)$ flavor symmetry which is enhanced to $SU(2) \times SU(2)$ in the infra-red (IR). By gauging the diagonal $SU(2)$ with non-zero Chern-Simons level $k$, we have the S-fold theory. The gauging breaks the $\mathcal{N} = 4$ SUSY to $\mathcal{N} = 3$ but the $\mathcal{N} = 4$ SUSY is restored in the IR and the theory flows to an non-trivial $\mathcal{N} = 4$ SCFT (resp. direct product of $\mathcal{N} = 4$ SCFTs) in the IR when $|k| \geq 4$ (resp. $|k| = 3$). Both Coulomb and Higgs branches of the $T[SU(2)]$ theory are lifted during the gauging and the S-fold SCFT has trivial Higgs and Coulomb branches and such a $\mathcal{N} = 4$ theory is called 3D $\mathcal{N} = 4$ rank-zero theory [6]. Refer to [11–17] for recent studies on the theory.

#### 2.1.2 Dual Abelian description

The S-fold SCFT $\mathcal{S}_k$ can be obtained from a twisted compactification of 6D (2,0) theory of $A_1$ type on a 3-manifold, once-punctured torus bundle with a monodromy element

$\varphi = ST^k \in SL(2, \mathbb{Z})$ [18]. Using the 3D-3D correspondence, we propose following dual description for the S-fold SCFT [19, 20],

$$
\mathcal{S}_k = \begin{cases} \text{3D } \mathcal{N} = 2\ U(1)_K^2 \text{ gauge theory coupled to two chirals } \Phi_{a=1,2}, & k = 3, \\ \left( U(1)_K^{r=k-1} \text{ coupled to } r \text{ chirals } \Phi_{a=1,\dots,r} \text{ with } \mathcal{W}_{\text{sup}} = \sum_{m=1}^{r-1} \mathcal{O}_m \right), & k \geq 4. \end{cases} \tag{2}
$$

See Appendix A for a detailed derivation. Here $\Phi_{1 \leq a \leq r}$ is a chiral multiplet whose charge under the $b$-th $U(1)$ gauge symmetry is $\delta_{ab} Q_a$ with

$$
Q_a = \begin{cases} 2, & a = 1, \\ 1, & 2 \leq a \leq r. \end{cases} \tag{3}
$$

The mixed Chern-Simons level $K$ for the $\mathcal{N} = 2\ U(1)^r$ gauge theory is given by[1]

$$
K = \begin{pmatrix} 2(k-2) & 0 & 2 & 4 & 6 & \dots & 2(k-3) \\ 0 & 2 & 2 & 2 & 2 & \dots & 2 \\ 2 & 2 & 4 & 4 & 4 & \dots & 4 \\ 4 & 2 & 4 & 6 & 6 & \dots & 6 \\ 6 & 2 & 4 & 6 & 8 & \dots & 8 \\ \dots & \dots & \dots & \dots & \dots & \dots \\ 2(k-3) & 2 & 4 & 6 & 8 & \dots & 2(k-2) \end{pmatrix}. \tag{4}
$$

$\mathcal{O}_m$'s in the superpotential $\mathcal{W}_{\text{sup}}$ are gauge-invariant 1/2 BPS monopole operators (more precisely, chiral primary multiplets containing the monopole operators):

$$
\begin{aligned}
\mathcal{O}_{m=1} &= \begin{cases} V_{(-1,-2,2)}, & k = 4, \\ V_{(-1,-1,\mathbf{0}_{k-5},-1,2)}, & k \geq 5, \end{cases} \\
\mathcal{O}_{m=2} &= V_{(0,2,-1,\mathbf{0}_{k-4})} \Phi_1, \\
\mathcal{O}_{m=3} &= V_{(0,-1,2,-1,\mathbf{0}_{k-5})}, \\
\mathcal{O}_{m=4} &= V_{(0,0,-1,2,-1,\mathbf{0}_{k-6})}, \\
&\dots \\
\mathcal{O}_{m=r-1} &= V_{(0,0,0,\dots,-1,2,-1)}.
\end{aligned} \tag{5}
$$

Here $\mathbf{0}_n = \overbrace{0,0,\dots,0}^{n}$ and $V_{\mathfrak{m}=(m_1,\dots,m_r)}$ denotes the 1/2 BPS bare monopole operator with flux $\mathfrak{m}$. The charge $q_a$ under the $a$-th $U(1)$ gauge symmetry of the monopole operator is

$$
q_a(V_{\mathfrak{m}}) = \sum_{b=1}^{r} \left( K_{ab} - \frac{1}{2} \delta_{ab} Q_a^2 \right) m_b - \frac{Q_a^2}{2} |m_a|, \tag{6}
$$

and one can confirm the gauge invariance of the superpotential $\mathcal{W}_{\text{sup}}$. The 2nd monopole operator $\mathcal{O}_{m=2}$ is also 1/2 BPS since it is purely electric or magnetic for each $U(1)$ factor in the gauge group.

The dual description has only manifest $\mathcal{N} = 2$ supersymmetry which is expected to be enhanced to $\mathcal{N} = 4$ in the IR. For $k \geq 4$, the superpotential deformation breaks the $U(1)^r$ topological symmetries (of $U(1)^r$ gauge symmetry) to $U(1)_A$ symmetry whose charge $A$ is

$$
A = (r-1) M_1 + M_2 + 2 M_3 + 3 M_4 + \dots + (r-1) M_r. \tag{7}
$$

---

[1]Here we use "$U(1)_{-1/2}$ quantization" of the chiral multiplet, i.e. we turn on background CS level $-1/2$ for the $U(1)$ symmetry of $\Phi$. In the usual convention, the (UV effective) mixed CS level is $K_{ab} - \frac{1}{2} \delta_{ab} Q_a^2$.

Here $M_a$ is the charge of $U(1)$ topological symmetry for $a$-th $U(1)$ gauge symmetry. The theory also has $U(1)_R$ R-symmetry which can be mixed with the $U(1)_A$ flavor symmetry. Let the R-charge at the general mixing be

$$R_\nu = R_{\nu=0} + \nu A, \tag{8}$$

with a mixing parameter $\nu \in \mathbb{R}$. We choose a reference R-charge $R_{\nu=0}$ to be the superconformal R-charge of the IR SCFT. Under the SUSY enhancement, the $U(1)_R \times U(1)_A$ is expected to be enhanced to $SO(4)_R \simeq SU(2)_C \times SU(2)_H$ R-symmetry with the following embedding

$$R_\nu = \left(J_3^C + J_3^H\right) + \nu\left(J_3^C - J_3^H\right). \tag{9}$$

Here $J_3^{C/H}$ denotes the Cartan generator of $SU(2)_{C/H}$ normalized as $J_3 \in \mathbb{Z}/2$.

**SUSY partition functions** The squashed 3-sphere partition function [21,22] of the $\mathcal{S}_k$ in (2) is (following the conventions in [23])

$$\mathcal{Z}_{S_b^3}(m,\nu) = \int \frac{d^r\mathbf{Z}}{(2\pi\hbar)^{\frac{r}{2}}} \mathcal{I}_\hbar(\mathbf{Z},m,\nu), \quad \text{where} \quad \mathbf{Z} = (Z_1, Z_2, \ldots, Z_r), \quad \text{and}$$

$$\mathcal{I}_\hbar(\mathbf{Z},m,\nu) = \exp\left(\frac{\mathbf{Z}^T K \mathbf{Z} + 2W\left((r-1)Z_1 + \sum_{a=1}^{r-1} aZ_{a+1}\right)}{2\hbar}\right) \prod_{a=1}^{r} \psi_\hbar(Q_a Z_a), \tag{10}$$

$$\text{with} \quad W = m + \left(i\pi + \frac{\hbar}{2}\right)(\nu - 1).$$

Here $\hbar = 2\pi i b^2$ with the squashing parameter $b$. $m$ is a rescaled real mass parameter, $b \times$ (real mass), of the $U(1)_A$ symmetry and $\nu$ is the R-symmetry mixing parameter in (9).[2] Using the following asymptotic expansion of the quantum dilogarithm function $\psi_{\hbar(Z)}$ [25],

$$\log\psi_\hbar(Z) \xrightarrow{\hbar\to 0} \frac{1}{\hbar}\mathrm{Li}_2(e^{-Z}) - \frac{1}{2}\log(1 - e^{-Z}) + O(\hbar), \tag{11}$$

the integrand can be expanded perturbatively in $\hbar$ as follows

$$\log\mathcal{I}(\mathbf{Z},m,\nu) \xrightarrow{\hbar\to 0} \frac{1}{\hbar}\mathcal{W}_0(\mathbf{Z},m,\nu) + \mathcal{W}_1(\mathbf{Z},m,\nu) + O(\hbar), \quad \text{where}$$

$$\mathcal{W}_0 = \sum_{a=1}^{r} \mathrm{Li}_2(e^{-Q_a Z_a}) + \frac{1}{2}\mathbf{Z}^T K \mathbf{Z} + (m + i\pi(\nu-1))\left((r-1)Z_1 + \sum_{a=1}^{r-1} aZ_{a+1}\right), \tag{12}$$

$$\mathcal{W}_1 = -\frac{1}{2}\sum_{a=1}^{r} \log(1 - e^{-Q_a Z_a}) + \frac{1}{2}(\nu-1)\left((r-1)Z_1 + \sum_{a=1}^{r-1} aZ_{a+1}\right).$$

Using the first two coefficients $\mathcal{W}_0$ and $\mathcal{W}_1$, the supersymmetric partition function $\mathcal{Z}_{(g,p)}$ on $\mathcal{M}_{g,p}$, degree $p$ bundle over genus $g$ Riemann surface $\Sigma_g$, with $p \in 2\mathbb{Z}$ can be computed in

---

[2]The overall additive constant of $\nu$ is fixed by requiring that the round 3-sphere partition function $|\mathcal{Z}_{S_{b=1}^3}(m=0,\nu)|$ is minimized at $\nu = 0$ [24]. Actually, the $|\mathcal{Z}_{S_{b=1}^3}(m=0,\nu)|$ is invariant under a sign change of $\nu$ which is related to the self-mirror property of the $T[SU(2)]$ and is related to the $\mathbb{Z}_2$ Weyl-symmetry, $M_A \leftrightarrow -M_A$, of the $SL(2,\mathbb{C})$ Chern-Simons theory in (A.4) through a 3D-3D relation (A.5).

the following way [23, 26, 27]

$$\mathcal{Z}_{(g,p)}(m,\nu) = \sum_{\mathbf{z}^\alpha \in \mathcal{S}_{\mathrm{B.E}}(m,\nu)} (H_\alpha(m,\nu))^{g-1} (F_\alpha(m,\nu))^p, \qquad \text{where}$$

$$\text{Bethe-vacua}: \mathcal{S}_{\mathrm{B.E}}(m,\nu) = \left\{ \mathbf{z} : \exp(\partial_{Z_b}\mathcal{W}_0)|_{\mathbf{Z}\to\log\mathbf{z}} = 1, \ b = 1,\dots,r \right\}$$

$$= \{\mathbf{z}^\alpha\}_{\alpha=0}^{|\mathcal{S}_{\mathrm{B.E}}|-1},$$

$$\text{Handle gluing operator}: H_\alpha(m,\nu) = \det_{a,b}\left(\partial_{Z_a}\partial_{Z_b}\mathcal{W}_0\right)e^{-2\mathcal{W}_1}\big|_{\mathbf{Z}\to\log\mathbf{z}^\alpha},$$

$$\text{Fibering operator}: F_\alpha(m,\nu) = \exp\left(-\frac{\mathcal{W}_0 - \sum_a Z_a \partial_{Z_a}\mathcal{W}_0 - m\partial_m\mathcal{W}_0}{2\pi i}\right)\Big|_{\mathbf{Z}\to\log\mathbf{z}^\alpha}. \tag{13}$$

For $p \in 2\mathbb{Z}$, there are two distinct supersymmetric backgrounds, $\nu_R = 0$ or $\nu_R = \frac{1}{2}$ in [27], depending on the spin-strucuture along the fiber $[S^1]$ direction. The above twisted partition function corresponds to the spin-strucuture with anti-periodic boundary condition ($\nu_R = \frac{1}{2}$). Especially when $p = 0$, the partition function can be identified with the twisted index $\mathcal{I}_g$ on a Riemann surface $\Sigma_g$ [28–31],

$$\mathcal{I}_g(\eta,\nu) := \mathrm{Tr}_{\mathcal{H}(\Sigma_g;\nu)}(-1)^{R_\nu}\eta^A = \mathcal{Z}_{(g,p=0)}(m = \log\eta, \nu). \tag{14}$$

We use the $U(1)_{R_\nu}$ R-symmetry for the topological twisting on $\Sigma_g$. Thus, the twisted index is well-defined only when the mixing parameter $\nu$ satisfies the following Dirac quantization condition

$$R_\nu(2-2g) \in \mathbb{Z}. \tag{15}$$

The twisted partition functions $\mathcal{Z}_{(g,p)}$ for $\mathcal{S}_k$ theory were computed in [6, 7] using the field theory description given in (1). One can check that they agree with the twisted partition functions given in (12) and (13). The non-trivial matches of various SUSY partition functions support the proposed IR duality between (1) and (2).

## 2.2 Generalized Haagerup TQFTs

In [6], they study the non-unitary semi-simple topological field theories (TQFTs), $\mathrm{TFT}_-[\mathcal{T}]$ and $\mathrm{TFT}_+[\mathcal{T}]$, associated to a 3D $\mathcal{N} = 4$ rank-0 SCFT $\mathcal{T}$. The two TQFTs are believed to be identical with the topologically twisted theories using $SU(2)_H$ and $SU(2)_C$ R-symmetry respectively. In 3D, $\mathcal{N} = 4$ (8 supercharges) is the minimal number of SUSY for a topological twisting. Being of rank-0 is important for the topologically twisted theory to be a genuine semi-simple TQFT which supports a rational chiral algebra on the boundary. For $\mathcal{N} = 4$ SCFTs of non-zero rank, the topologically twisted theories are non-semisimple and the corresponding 2D chiral algebras are generically logarithmic instead of rational [32, 33]. Unitarity is broken in the topological twisting procedure. From all these considerations, topologically twisted $\mathcal{N} = 4$ rank-0 SCFTs seem to provide a natural physical realization of non-unitary semi-simple TQFTs.

There have been several previous works realizing non-unitary TQFTs in physical (2+1)D systems. In [34], they obtained non-unitary TQFTs from M5 branes wrapped on some non-hyperbolic 3-manifolds. Later it is found that all such systems flow to 3D rank-zero SCFTs in the IR [35]. In [36], they consider 3D $\mathcal{N} = 2$ gauge theories whose half-indices give the characters of non-unitary minimal models. The gauge theories also turned out to experience non-trivial SUSY enhancement and flow to 3D rank-0 SCFTs in the IR. So these realizations are actually equivalent to ours. In [37], they realize non-unitary TQFTs from a non-trivial $S^1$-reduction of 4D Argyres-Douglas theories. There is also an attempt to understand the bulk dual of a non-unitary minimal model $M(3,5)$ using a candidate bulk ground state wave function called

Table 1: Basic dictionaries of (rank-0 SCFT)/(non-unitary TQFTs) correspondence. See more dictionaries in [6].

| $\text{TFT}_\pm[\mathcal{T}]$ | Rank-0 SCFT $\mathcal{T}$ |
|---|---|
| $\alpha$ : simple object | Bethe-vacuum $\mathbf{z}^\alpha \in \mathcal{S}_{\text{B.E}}(m=0, \nu=\pm 1)$ |
| $S_{0\alpha}^2$ of $\text{TFT}_\pm$ | $H_\alpha^{-1}(m=0, \nu=\pm 1)$ |
| $\frac{T_{\alpha\alpha}}{T_{00}}$ of $\text{TFT}_\pm$ | $\frac{F_\alpha(m=0,\nu=\pm1)}{F_{\alpha=0}(m=0,\nu=\pm1)}$ |

Gaffnian state [38]. We need further study to check whether these realizations are eventually equivalent to ours or not.

In the (rank-0 SCFT)/(non-unitary TQFTs) correspondence, it is claimed that[3]

$$\left( \mathcal{Z}_{(g,p)}(m=0, \nu=\pm 1) \text{ of } \mathcal{T} \right) = \mathcal{Z}\left[ \text{TFT}_\pm[\mathcal{T}] \text{ on } \mathcal{M}_{g,p} \right]. \tag{16}$$

According to the claim, taking the degenerate limit, $(m, \nu) \to (0, +1)$ or $(m, \nu) \to (0, -1)$, on the BPS partition functions is somehow equivalent to the procedure of topological twisting, using $SU(2)_C$ or $SU(2)_H$ R-symmetry. The equivalence is manifest for the SUSY backgrounds with $p = 0$ but is not quite obvious for general $\mathcal{Z}_{(g,p)}$. Notice that the Dirac quantization condition (15) is automatically met in the degenerate limits since

$$R_{\nu=1} = 2J_3^C \in \mathbb{Z} \text{ and } R_{\nu=-1} = 2J_3^H \in \mathbb{Z}. \tag{17}$$

Combining the localization result for $\mathcal{Z}_{(g,p)}$ in (13) with the following general formula for (2+1)D bosonic TQFT

$$\mathcal{Z}[\text{TQFT on } \mathcal{M}_{g,p}] = \sum_{\alpha\,:\,\text{simple objects}} (S_{0\alpha})^{2-2g} (T_{\alpha\alpha})^p \,,$$

we have the basic dictionaries in Table 1. In the table, $(S_{\alpha\beta}, T_{\alpha\beta})$ are modular matrices and the simple object $\alpha = 0$ corresponds to the trivial one. Among Bethe-vacua in $\mathcal{S}_{\text{B.E}}$, the one $\mathbf{z}^{\alpha=0}$ corresponding to the trivial simple object is chosen to satisfy the following consistency condition

$$\left| \sum_\alpha H_\alpha^{-1} F_\alpha \right| = \left| \sum_\alpha S_{0\alpha}^2 T_{\alpha\alpha} \right| = |(STS)_{00}| = |(T^{-1}ST^{-1})_{00}| = |S_{00}| = |H_{\alpha=0}|^{-1/2}. \tag{18}$$

Here we use the $SL(2, \mathbb{Z})$ relations $S^2 = (ST)^3 = \mathcal{C}$ where the charge conjugation $\mathcal{C}$ is an identity matrix in our case. Using the relations in the above table and (18), one can obtain partial information on the modular data of the non-unitary TQFTs, $\text{TFT}_\pm$. Imposing general consistency conditions on the top of the partial information, we obtain following consistent

---

[3]The two partition functions have subtle overall phase factors depending on 3-manifold framing, background Chern-Simons level of $U(1)_R$ symmetry and etc. We will not keep track of all these subtle choices and the equality should be understood as an equality up to an overall phase factor.

modular data $(k \geq 3)$ [7][4]

$(S_{\alpha\beta}$ of TFT$_-[\mathcal{S}_k])$

$$
= \left(
\begin{array}{cc|cc|ccccc|cccc}
(-1)^k a_0 \, a_0 & a_3 \,(-1)^k a_3 & -a_1 \, a_1 -a_1 \cdots (-1)^{k-3} a_1 & a_2 \,-a_2 \, a_2 \cdots (-1)^{k+2} a_2 \\
a_0 \quad a_0 & a_3 \quad a_3 & a_1 \; a_1 \; a_1 \; \cdots \quad a_1 & -a_2 -a_2 -a_2 \cdots \quad -a_2 \\
\hline
a_3 \quad a_3 & a_0 \quad a_0 & a_1 \; a_1 \; a_1 \; \cdots \quad a_1 & a_2 \; a_2 \; a_2 \; \cdots \quad a_2 \\
(-1)^k a_3 \, a_3 & a_0 \,(-1)^k a_0 & -a_1 \, a_1 -a_1 \cdots (-1)^{k-3} a_1 & -a_2 \, a_2 -a_2 \cdots (-1)^{k+1} a_2 \\
\hline
\begin{array}{cc} -a_1 & a_1 \\ a_1 & a_1 \\ -a_1 & a_1 \\ \vdots & \vdots \\ (-1)^{k-3} a_1 & a_1 \end{array}
\begin{array}{cc} a_1 & -a_1 \\ a_1 & a_1 \\ a_1 & -a_1 \\ \vdots & \vdots \\ a_1 & (-1)^{k-3} a_1 \end{array}
& & 2a_1 \cos \dfrac{ij\pi}{k-2}\Big|_{1\leq i,j \leq k-3} & 0 \\
\hline
\begin{array}{cc} a_2 & -a_2 \\ -a_2 & -a_2 \\ a_2 & -a_2 \\ \vdots & \vdots \\ (-1)^{k+2} a_2 & -a_2 \end{array}
\begin{array}{cc} a_2 & -a_2 \\ a_2 & a_2 \\ a_2 & -a_2 \\ \vdots & \vdots \\ a_2 & (-1)^{k+1} a_2 \end{array}
& & 0 & 2a_2 \cos \dfrac{ij\pi}{k+2}\Big|_{1\leq i,j \leq k+1}
\end{array}
\right),
$$

$(T_{\alpha\beta}$ of TFT$_-[\mathcal{S}_k]) = \delta_{\alpha\beta} \exp\left( 2\pi i \left( h_\alpha - \dfrac{c}{24} \right) \right),$   with

$$
\vec{h} = \left\{ \frac{k+2}{4}, 0, 0, \frac{k+2}{4}, \frac{A^2}{4(k-2)}\Big|_{A=1,\ldots,k-3}, \frac{B^2}{4(k+2)}\Big|_{B=1,\ldots,k+1} \right\} - \frac{k+2}{4} \pmod 1, \quad \text{and}
$$

$$
c = -(6k + 11) \pmod{24}. \tag{19}
$$

They are square matrices of dimension $(2k+2)$. Here we introduce

$$
(a_0, a_1, a_2, a_3) = \left( \frac{1}{\sqrt{8(k-2)}} + \frac{1}{\sqrt{8(k+2)}}, \frac{1}{\sqrt{2(k-2)}}, \frac{1}{\sqrt{2(k+2)}}, \frac{1}{\sqrt{8(k-2)}} - \frac{1}{\sqrt{8(k+2)}} \right). \tag{20}
$$

**Bethe-vacuum to loop operator map**   Generally, the Hilbert-space of a bosonic 3D TQFT on a two-torus $\mathbb{T}^2$ can be given as follows

$$
\mathcal{H}(\mathbb{T}^2) = \text{Span}\{|\alpha\rangle \,:\, \alpha = 0, 1, \ldots, n-1\}, \quad \text{with} \quad |\alpha\rangle = \hat{\mathcal{L}}^\alpha_{(0,1)}|0\rangle. \tag{21}
$$

For our case, i.e. when the TQFT is TFT$_-[\mathcal{S}_k]$, the state $|\alpha\rangle$ can be identified with the Bethe-vacuum $\mathbf{z}^\alpha \in \mathcal{S}_{\text{B.E}}(m=0, \nu=-1)$ in (13). The trivial vacuum $|\alpha = 0\rangle$ is identified with the Bethe-vacuum $\mathbf{z}^{\alpha=0}$ satisfying the relation in (18). Here $\hat{\mathcal{L}}^\alpha_{(p,q)}$ is the quantum loop operator of anyon type $\alpha$ supported on a $(p,q)$-cycle on the two-torus. From the analysis, we expect that there is a natural one-to-one map between the Bethe-vacua and the types of loop operators. We claim that the first four simple objects (or Bethe-vacua) in TFT$_-[\mathcal{S}_k]$ correspond to following loop operators

$\alpha = 0 :$  Trivial loop operator $\mathbf{1}$,

$\alpha = 1 :$  Topological defect $\mathcal{L}^{\mathbb{Z}_2}$ associated to the $\mathbb{Z}_2$ 1-form symmetry,

$\alpha = 2 :$  SUSY Wilson loop $\mathcal{L}^W$ with gauge charge $\mathbf{Q}_W = (r-1, 1, 2, \ldots, r-1)$,

$\alpha = 3 :$  $\mathcal{L}^{\mathbb{Z}_2} * \mathcal{L}^W$ .

$\qquad\qquad\qquad\qquad\qquad\qquad\qquad\qquad\qquad\qquad\qquad\qquad\qquad\qquad$ (22)

---

[4]The modular data presented here is slightly different from the one in [7]. We exchanged $\alpha = 0 \leftrightarrow \alpha = 1$ and permutated other $\alpha$'s from the previous one. Generally, an exchange of the vacuum $\alpha = 0$ and $\alpha \neq 0$ could give an inconsistent modular data with some negative fusion coefficients. In our case, however, both are consistent modular data. The two consistent choices of modular data are related to the ambiguity of choosing the $\mathbf{z}^{\alpha=0}$ from the relation in (18) due to the fact that $|H_{\alpha=0}| = |H_{\alpha=1}|$. At the level of RCFT characters, two choices are simply related to each other by exchanging $\chi_{\alpha=0} \leftrightarrow \chi_{\alpha=1}$ and permutating other $\chi_\alpha$'s accordingly.

The map for $\alpha = 0$ is obvious. The modular data has $\mathbb{Z}_2$ 1-form symmetry generated by an anyon $\alpha = 1$. The symmetry is originated from the $\mathbb{Z}_2$ center subgroup of $SU(2)$ gauge theory in (1) or from the $\mathbb{Z}_2 = \{\pm 1\}$ subgroup of the first $U(1)$ gauge group in the dual Abelian theory (2). The 1-form symmetry defines a codimension 2 topological defect, which is the $\mathcal{L}^{\mathbb{Z}_2}$. To understand the map for $\alpha = 2$, consider the action of loop operators $\hat{\mathcal{L}}^{\alpha}_{(1,0)}$ on the basis

$$\hat{\mathcal{L}}^{\beta}_{(1,0)} |\alpha\rangle = \frac{S_{\beta\alpha}}{S_{0\alpha}} |\alpha\rangle \,. \tag{23}$$

The supersymmetric Wilson loop $\hat{\mathcal{L}}^{W}_{(1,0)}$ with gauge charge $\mathbf{Q} = (Q_1, \ldots, Q_r)$ acts on the Bethe-vacua as follows [31]

$$\hat{\mathcal{L}}^{W}_{(1,0)} |\alpha\rangle = \left( \prod_{a=1}^{r} z_a^{Q_a} |_{\mathbf{z} \to \mathbf{z}^{\alpha}} \right) |\alpha\rangle \,. \tag{24}$$

For the case when $\mathbf{Q} = \mathbf{Q}_W = (r-1, 1, 2, \ldots, r-1)$, one can check that

$$\left( \prod_{a=1}^{r} z_a^{Q_a} |_{\mathbf{z} \to \mathbf{z}^{\alpha}} \right) = \left( \frac{S_{\beta\alpha}}{S_{0\alpha}} \text{ with } \beta = 2 \right) \,. \tag{25}$$

From (23), (24) and (25), we have the map for $\alpha = 2$. Finally, the map for $\alpha = 3$ follows from the fusion rule $\mathcal{L}^{\alpha=1} * \mathcal{L}^{\alpha=2} = \mathcal{L}^{\alpha=3}$ which can be read off from the S-matrix using Verlinde's formula. We also tried to find the map for other Bethe-vacua but were not successful in several attempts.

**Gauging the $\mathbb{Z}_2$ 1-form symmetry**   The $\mathbb{Z}_2$ 1-form symmetry has non-trivial 't Hooft anomaly for odd $k$. It follows from the fact that the topological spin $h_{\alpha}$ of the symmetry generating anyon $\alpha = 1$ is $\pm \frac{1}{4}$ (mod 1) instead of 0 or $\frac{1}{2}$ [39]. The 1-form symmetry is non-anomalous for $k \in 2\mathbb{Z}$. In the Abelian UV description in (2), gauging the 1-form symmetry is equivalent to rescaling the first $U(1)$ gauge field $A$ to $\frac{A}{2}$. Then, the mixed CS level becomes $\frac{K_{ab}}{Q_b Q_b}$ which is not properly quantized for odd $k$. It means that there is an obstruction of the gauging, i.e. a 't Hooft anomaly, for odd $k$. The 't Hooft anomay in the UV description (1) using $T[SU(2)]$ theory was found in [40]. Using the $\mathbb{Z}_2$ 1-form symmetry, we define the $\mathbb{Z}_2$ gauged S-fold theory $\widetilde{\mathcal{S}}_k$ as

$$\widetilde{\mathcal{S}}_k := \begin{cases} \frac{\mathcal{S}_k}{\mathbb{Z}_2} \,, & k \in 2\mathbb{Z} \,, \\ \frac{\mathcal{S}_k \otimes \mathcal{A}^{\pm}_{\mathbb{Z}_2}}{\mathbb{Z}_2^{\text{diag}}} \,, & k \in 4\mathbb{Z} \pm 1 \,. \end{cases} \tag{26}$$

Here $/\mathbb{Z}_2$ denotes gauging the $\mathbb{Z}_2$ 1-form symmetry. Here $\mathcal{A}^{\pm}_{\mathbb{Z}_2}$ is a topological field theory with anomalous $\mathbb{Z}_2$ 1-form symmetry generated by an anyon with topological spin $\pm \frac{1}{4}$. Then, the diagonal $\mathbb{Z}_2$ 1-form symmetry is non-anomalous and thus can be gauged. The $\mathcal{A}^{\pm}_{\mathbb{Z}_2}$ is expected to have two simple objects, $\alpha = 0$ and $\alpha = 1$, related to each other by the $\mathbb{Z}_2$ 1-form symmetry and has following S-matrix

$$\left( S\text{-matrix of } \mathcal{A}^{\pm}_{\mathbb{Z}_2} \right) = \frac{1}{\sqrt{2}} \begin{pmatrix} 1 & 1 \\ 1 & -1 \end{pmatrix} \,. \tag{27}$$

Let $\text{TFT}_-[\widetilde{\mathcal{S}}_k]$ be the non-unitary TQFT associated to the $\widetilde{\mathcal{S}}_k$:

$$\text{TFT}_-[\widetilde{\mathcal{S}}_k] := \begin{cases} \frac{\text{TFT}_-[\mathcal{S}_k]}{\mathbb{Z}_2} \,, & k \in 2\mathbb{Z} \,, \\ \frac{\text{TFT}_-[\mathcal{S}_k] \otimes \mathcal{A}^{\pm}_{\mathbb{Z}_2}}{\mathbb{Z}_2^{\text{diag}}} \,, & k \in 4\mathbb{Z} \pm 1 \,. \end{cases} \tag{28}$$

The gauged $\mathbb{Z}_2$ 1-form symmetry is fermionic for $k \in 4\mathbb{Z}$ and bosonic otherwise. Thus, the topological theory TFT$_-[\widetilde{\mathcal{S}}_k]$ after the gauging is fermionic (i.e. spin TQFT) when $k \in 4\mathbb{Z}$ and bosonic (i.e. non-spin TQFT) otherwise. The modular data, $\widetilde{S}$ and $\widetilde{T}$, for the bosonic TQFT TFT$_-[\widetilde{\mathcal{S}}_k]$ is given as follows [7]

For $\quad k \in 4\mathbb{Z}_{\geq 1} \pm 1$,

$$
\widetilde{S} = \sqrt{2}
\left(
\begin{array}{cc|ccc|ccc}
a_0 & a_3 & a_1 & \cdots & a_1 & -a_2 & \cdots & -a_2 \\
a_3 & a_0 & a_1 & \cdots & a_1 & a_2 & \cdots & a_2 \\
\hline
a_1 & a_1 & & & & & & \\
\vdots & \vdots & & 2a_1 \cos \frac{4ij\pi}{k-2}\Big|_{1\leq i,j \leq \frac{k-3}{2}} & & & 0 & \\
a_1 & a_1 & & & & & & \\
\hline
-a_2 & a_2 & & & & & & \\
\vdots & \vdots & & 0 & & & 2a_2 \cos \frac{4ij\pi}{k+2}\Big|_{1\leq i,j \leq \frac{k+1}{2}} & \\
-a_2 & a_2 & & & & & &
\end{array}
\right),
$$
(29)

$$
\widetilde{T} = e^{-\frac{2\pi i c}{24}} \operatorname{diag}\left[ \exp\left( 2\pi i \left\{ 0,0, \frac{A^2}{(k-2)}\Big|_{A=1,\cdots,\frac{k-3}{2}}, \frac{B^2}{(k+2)}\Big|_{B=1,\cdots,\frac{k+1}{2}} \right\} \right) \right],
$$

$$
c = (1 \pm 1)(\bmod 8).
$$

They are square matrices of dimension $(k+1)$.

For $k \in 4\mathbb{Z}_{\geq 1} + 2$,

$$
\widetilde{S} =
\left(
\begin{array}{cc|ccc|cc|ccc|cc}
2a_0 & 2a_3 & 2a_1 & \cdots & 2a_1 & a_1 & a_1 & -2a_2 & \cdots & -2a_2 & -a_2 & -a_2 \\
2a_3 & 2a_0 & 2a_1 & \cdots & 2a_1 & a_1 & a_1 & 2a_2 & \cdots & 2a_2 & a_2 & a_2 \\
\hline
2a_1 & 2a_1 & & & & & & & & & & \\
\vdots & \vdots & & 4a_1 \cos \frac{ij\pi}{n}\Big|_{1\leq i,j\leq n-1} & & 2a_1\mathbb{J}_{n-1} & 2a_1\mathbb{J}_{n-1} & & 0 & & & 0 \\
2a_1 & 2a_1 & & & & & & & & & & \\
\hline
a_1 & a_1 & & 2a_1\mathbb{J}_{n-1}^T & & b_{1,+} & b_{1,-} & & & & \frac{i^n}{2\sqrt{2}} & -\frac{i^n}{2\sqrt{2}} \\
a_1 & a_1 & & 2a_1\mathbb{J}_{n-1}^T & & b_{1,-} & b_{1,+} & & 0 & & -\frac{i^n}{2\sqrt{2}} & \frac{i^n}{2\sqrt{2}} \\
\hline
-2a_2 & 2a_2 & & & & & & & & & & \\
\vdots & \vdots & & 0 & & & 0 & & 4a_2 \cos \frac{ij\pi}{n+1}\Big|_{1\leq i,j\leq n} & & 2a_2\mathbb{J}_n & 2a_2\mathbb{J}_n \\
-2a_2 & 2a_2 & & & & & & & & & & \\
\hline
-a_2 & a_2 & & 0 & & \frac{i^n}{2\sqrt{2}} & -\frac{i^n}{2\sqrt{2}} & & 2a_2\mathbb{J}_n^T & & b_{2,+} & b_{2,-} \\
-a_2 & a_2 & & & & -\frac{i^n}{2\sqrt{2}} & \frac{i^n}{2\sqrt{2}} & & 2a_2\mathbb{J}_n^T & & b_{2,-} & b_{2,+}
\end{array}
\right),
$$

$$
\widetilde{T} = e^{-\frac{2\pi i}{24}} \operatorname{diag}\left[ \exp\left( 2\pi i \left\{ 0,0, \frac{A^2}{(4n)}\Big|_{A=1,\cdots,n-1}, \frac{n}{4}^{\otimes 2}, \frac{B^2}{4(n+1)}\Big|_{B=1,\cdots,n}, \frac{n+1}{4}^{\otimes 2} \right\} \right) \right].
$$
(30)

They are square matrices of dimension $(\frac{k+8}{2})$. Here we define

$$
n = \frac{k-2}{4}, \qquad \mathbb{J}_n^T = (-1, 1, -1, \cdots, (-1)^n),
$$

$$
b_{1,\pm} = (-1)^n a_1 \pm \frac{i^n}{2\sqrt{2}}, \quad \text{and} \quad b_{2,\pm} = (-1)^{n+1} a_2 \pm \frac{(-i)^n}{2\sqrt{2}}.
$$
(31)

For $k = 4m^2 + 4m + 3$ $(m \in \mathbb{Z}_{\geq 0})$, the modular data, $(\widetilde{S}, \widetilde{T})$, is related to the generalized Haagerup-Izumi modular data $\mathcal{D}^{\omega=2}\mathrm{Hg}_{2m+1}$ [41] by a Hecke (or Galois conjugate) transformation. The specific form of Galois conjugation [42, 43] is given by

$$
\begin{aligned}
\left( S \text{ of } \mathcal{D}^{\omega=2}\mathrm{Hg}_{2m+1} \text{ in } [41] \right) &\cong \widetilde{T}^{\bar p} \widetilde{S}^{-1} \widetilde{T}^p \widetilde{S} \widetilde{T}^{\bar p} \widetilde{S}^2, \\
\left( T \text{ of } \mathcal{D}^{\omega=2}\mathrm{Hg}_{2m+1} \text{ in } [41] \right) &\cong \widetilde{T}^{\bar p},
\end{aligned}
$$
(32)

up to permutations, for $p = 2$ and $\bar{p} = \frac{k^2-3}{2}$. When $k = 4m^2 + 4m + 3$, the conductor $N$ of the modular matrices, i.e. the minimal positive integer such that $T^N = 1$, is $N = (k+2)(k-2)$ and $\bar{p}$ is a multiplcative inverse of $p$ in $\mathbb{Z}/N\mathbb{Z}$. Especially when $m = 1$, the $\mathcal{D}^{\omega=0}\mathrm{Hg}_{2m+1}$ is the modular data of original Haagerup TQFT with 12 simple objects. The Haggerup TQFT has drawn much attentions since it serve as an example of 3D extoic unitary TQFTs, which can not be realized as a Chern-Simons theory or its discrete gauging. The modular data $\mathcal{D}^{\omega}\mathrm{Hg}_{2m+1}$ provides a two-parameter generalization of the Haagerup TQFT's modular data, parametrized by $m \geq 0$ and $\omega \in \mathbb{Z}/((2m+1)\mathbb{Z})$. Our $\mathrm{TFT}_-[\mathcal{S}_k]$ and $\mathrm{TFT}_-[\widetilde{\mathcal{S}}_k]$ can be regarded as (non-unitary) generalized Haagerup TQFTs. The modular data of $\mathrm{TFT}_-[\widetilde{\mathcal{S}}_k]$ for $k = 8$ is equivalent to one of rank 10 fermionic modular data found in [44].

# 3 Generalized non-unitary Haagerup RCFTs and their characters

Let us consider the Haagerup TQFT on a solid torus with a consistent holomorphic boundary condition $\mathcal{B}$. The solid torus is equivalently $D_2 \times S^1$ where $D_2$ is a disk and $\partial D_2 = \widetilde{S}^1$. A TQFT equipped with MTC has various loop operators associated with the anyons. We insert a loop operator $\mathcal{L}^{\alpha}$ running around the circle $S^1$ at the center of $D_2$. Upon the boundary condition, let us denote by $\{\chi_{\alpha}(\tau)\}$ the expectation value of the loop operators. Here $\chi_0(\tau)$ is the partition function on $D_2 \times S^1$ with no loop operator insertion, and $\tau$ is the complex modulus of the boundary two-torus $T^2 = S^1 \times \widetilde{S}^1$. Given the modular data $S$ and $T$ (19) in the bulk, one can argue that $\{\chi_{\alpha}(\tau)\}$ should obey the transformation rules below,

$$
\begin{aligned}
\chi_{\alpha}(\tau + 1) &= \sum_{\beta} T_{\alpha\beta} \chi_{\beta}(\tau), \\
\chi_{\alpha}(-1/\tau) &= \sum_{\beta} S_{\alpha\beta} \chi_{\beta}(\tau).
\end{aligned}
\tag{33}
$$

In other words, $\{\chi_{\alpha}(\tau)\}$, which is referred to as the partition vectors, form a vector-valued modular form under the $SL(2, \mathbb{Z})$.

On the boundary of TQFT, there exists a chiral current that defines a certain chiral algebra. Each partition vector $\chi_{\alpha}(\tau)$ can be often understood as the character of the chiral algebra. More precisely, the Hilbert space of the TQFT on $D_2$ becomes the vacuum representation of the chiral algebra. When a loop operator $\mathcal{L}^{\alpha}$ is placed at the center of $D_2$, the quantization then leads to the equivalence between the Hilbert space and a highest weight representation of the chiral algebra other than the vacuum. As a remark, this chirality is reflected in the the holomorphic dependence of $\{\chi_{\alpha}(\tau)\}$ on the complex modulus $\tau$.

One expects that a chiral algebra living on the boundary includes the Virasoro algebra, and $\{\chi_{\alpha}(\tau)\}$ can be identified as the conformal characters of a putative rational CFT (RCFT). Indeed, the Haagerup TQFT is likely to have the Virasoro algebra on the boundary. This is because the holomorphic twist of the 3D supersymmetric theories gives a stress tensor on the boundary when the bulk theory is topological [45]. It is therefore interesting to see if the candidate characters $\{\chi_{\alpha}(\tau)\}$ obtained from the Haagerup TQFT can be regarded as the conformal characters of 2D RCFTs. We will refer them as to the Haagerup RCFTs, denoted by $\mathcal{R}_k$.

Obviously, the consistent modular transformation of such candidate conformal characters $\{\chi_{\alpha}(\tau)\}$ does not guarantee that the putative RCFT exists. What else should $\{\chi_{\alpha}(\tau)\}$ satisfy to convince that the Haagerup RCFTs exist?

In the limit $\tau \to i\infty$, each candidate conformal character $\chi_\alpha(\tau)$ can be expanded in powers of $q = e^{2\pi i \tau}$ as

$$\chi_\alpha(q) = q^{h_\alpha - \frac{c}{24}} \sum_n a_\alpha(n) q^n \,, \tag{34}$$

where the leading exponent can be identified as the central charge of the putative RCFT and the conformal weight $h_\alpha$ of the chiral primary. One can also argue that the conformal weight of the chiral primary equals to the statistical spin of the loop operator in the bulk modulo integer. The salient feature of conformal characters of a consistent rational CFT is that each Fourier coefficient $a_\alpha(n)$ in (34) is non-negative integer-valued. This is because $a_\alpha(n)$ counts the number of states of given conformal weight.

The integrality of Fourier coefficients $a_\alpha(n)$ requires that the set of candidate characters $\chi_\alpha(q)$ is a representation of $SL(2, \mathbb{Z}_N)$ for a certain $N$, which is known as the integrality theorem [46, 47] or the congruence property in the modular tensor category [42]. The integrality theorem plays a key role in classifying the space of bosonic RCFTs [48], and it has been further generalized to the fermionic theories recently [49, 50]. As expected, one can show that the conformal weights together with modular matrices (19) are indeed consistent to a reducible representation of $SL(2, \mathbb{Z}_N)$. However, the non-negative coefficients are not guaranteed yet.

The present work aims to explicitly construct the conformal characters that transform as a vector-valued modular form (vvmf) with the Haagerup modular data (19) and have non-negative integer Fourier coefficients. Given such conformal characters, we strongly suspect that the Haagerup RCFTs $\mathcal{R}_k$ actually exist. Their existence also implies that the Haagreup TQFTs have consistent modular tensor categories beyond the mere modular data. In fact, one can view the full 2D Haagerup RCFTs $\mathcal{R}_k$ as the Haagerup TQFTs $\text{TFT}_-[\mathcal{S}_k]$ on an interval.

## 3.1 UV Abelian gauge theory and the half-index

We can utilize the aforementioned UV Abelian gauge theory (2) leading to the Haagerup TQFT to obtain the explicit $q$-expansion of a few candidate conformal characters.

To do so, we first need to translate the consistent holomorphic boundary condition $\mathcal{B}$ of the TQFT to a supersymmetric boundary condition in the UV Abelian gauge theory $\mathcal{S}_k$ on $D_2 \times S^1$. We propose that[5] the boundary condition $\mathcal{B}$ in the IR can be described by a simple SUSY boundary condition in $\mathcal{S}_k$, which sets any chiral multiplet $\{\phi, \psi_\pm, F\}$ to a deformed Dirichlet boundary condition ($D_c$),

$$\phi|_\partial = c \,, \qquad \psi_+|_\partial = 0 \,, \tag{35}$$

with a non-zero constant $c$, and any vector multiplet $\{A_\mu, \lambda_\pm, \sigma, D\}$ to a Dirichlet boundary condition ($\mathcal{D}$),

$$A_0 \pm A_1|_\partial = 0 \,, \qquad D|_\partial = 0 \,, \qquad \lambda_-|_\partial = 0 \,. \tag{36}$$

Given the above SUSY boundary condition (35) and (36), denoted by $\mathcal{B}' = (D_c, \mathcal{D})$ collectively, the half-index can be defined as [51–54]

$$\mathcal{I}_{\mathcal{B}'}(q, \eta, \nu) = \text{Tr}_{\mathcal{H}} \left[ (-1)^{R_\nu} q^{\frac{R_\nu}{2} + j_3} \eta^A \right] \,, \tag{37}$$

where $\mathcal{H}$ is the Hilbert space of a given SCFT on the disk, and the circle $S^1$ can be understood as the temporal circle. Specifically, the half-index of the SCFT $\mathcal{S}_k$ (2) is given by [54]

$$\mathcal{I}_{\mathcal{B}'}(q, \eta, \nu) = \frac{1}{(q)_\infty^r} \sum_{\mathfrak{m}} q^{\frac{1}{2}\mathfrak{m}^T \cdot K \cdot \mathfrak{m}} \left[ (-q^{\frac{1}{2}})^{\nu-1} \eta \right]^{(r-1)m_1 + \sum_{a=1}^{r}(a-1)m_a} \prod_{a=1}^{r} (q^{-Q_a m_a + 1}; q)_\infty \,, \tag{38}$$

---

[5]Verifying whether the UV boundary condition flows to an IR SCFT boundary condition compatible with the topological twisting is a highly non-trivial task. Consistent RCFT characters are obtained, as we will see below, from the half-indices associated with the boundary condition $\mathcal{B}' = (D_c, \mathcal{D})$. This may imply that $\mathcal{B}'$ is compatible with the topological twisting.

where the magnetic fluxes $\mathfrak{m} = (m_1, m_2, \ldots, m_r)$ are quantized, i.e., $m_a \in \mathbb{Z}$ for all $a$. The $K$ matrix and the gauge charges $Q_a$ are given by (4) and (3). We remark that the R-charges of chiral multiplets should be zero. Otherwise, the R-charge would not be compatible with (35). The Pochhammer symbols are defined by

$$(x; q)_k := \prod_{n=0}^{k-1}(1 - q^n x), \qquad (q)_k := (q; q)_k := \prod_{n=1}^{k}(1 - q^n).\tag{39}$$

One can further simplify the half-index (38) into

$$\mathcal{I}_{\mathcal{B}'}(q, \eta, \nu) = \sum_{m_a \in \mathbb{Z}_{\geq 0}} \frac{q^{\frac{1}{2}\mathfrak{m}^T \cdot K \cdot \mathfrak{m}}}{(q)_{2m_1}(q)_{m_2} \ldots (q)_{m_r}}\left[(-q^{\frac{1}{2}})^{\nu-1}\eta\right]^{-(r-1)m_1 - \sum_{a=1}^{r}(a-1)m_a}.\tag{40}$$

Some of supersymmetric loop operators $\mathcal{L}^\alpha$ (22) in the SCFT $\mathcal{S}_k$ flow to the loop operators in the Haagerup TQFT $\mathrm{TFT}_-[\mathcal{S}_k]$. The insertion of $\mathcal{L}^\alpha(S^1)$ placed at the center of $D_2$ corresponds to adding to $\mathcal{S}_k$ a certain static source at the center. In the presence of $\mathcal{L}^\alpha(S^1)$, the half-index thus becomes

$$\mathcal{I}_{\mathcal{B}'}^\alpha(q, \eta, \nu) = \mathrm{Tr}_{\mathcal{H}^\alpha}\left[(-1)^{R_\nu}q^{\frac{R_\nu}{2}+j_3}\eta^A\right],\tag{41}$$

where $\mathcal{H}^\alpha$ denotes the Hilbert space of $\mathcal{S}_k$ on $D_2$ with the source. Using the SUSY localization technique, one can compute the half-indices (41) exactly:

$$\mathcal{I}_{\mathcal{B}'}^{\alpha=1}(q, \eta, \nu) = \sum_{\mathfrak{m} \in \mathcal{Z}'} \frac{q^{\frac{1}{2}\mathfrak{m}^T \cdot K \cdot \mathfrak{m}}}{(q)_{2m_1}(q)_{m_2} \ldots (q)_{m_r}}\left[(-q^{\frac{1}{2}})^{\nu-1}\eta\right]^{-(r-1)m_1 - \sum_{a=1}^{r}(a-1)m_a},$$

$$\mathcal{I}_{\mathcal{B}'}^{\alpha=2}(q, \eta, \nu) = \sum_{\mathfrak{m} \in \mathcal{Z}} \frac{q^{\frac{1}{2}\mathfrak{m}^T \cdot K \cdot \mathfrak{m}}}{(q)_{2m_1}(q)_{m_2} \ldots (q)_{m_r}}\left[q(-q^{\frac{1}{2}})^{\nu-1}\eta\right]^{-(r-1)m_1 - \sum_{a=1}^{r}(a-1)m_a},\tag{42}$$

$$\mathcal{I}_{\mathcal{B}'}^{\alpha=3}(q, \eta, \nu) = \sum_{\mathfrak{m} \in \mathcal{Z}'} \frac{q^{\frac{1}{2}\mathfrak{m}^T \cdot K \cdot \mathfrak{m}}}{(q)_{2m_1}(q)_{m_2} \ldots (q)_{m_r}}\left[q(-q^{\frac{1}{2}})^{\nu-1}\eta\right]^{-(r-1)m_1 - \sum_{a=1}^{r}(a-1)m_a}.$$

Here $\mathcal{Z}$ and $\mathcal{Z}'$ are defined as

$$\mathcal{Z} = \left(\mathbb{Z}_{\geq 0}\right)^r, \qquad \mathcal{Z}' = \left(\mathbb{Z}_{\geq 0} + \frac{1}{2}\right) \otimes \left(\mathbb{Z}_{\geq 0}\right)^{r-1}.\tag{43}$$

The half-integer values of the $U(1)$ flux $m_1$ can be explained by the fact that the static sources corresponding to $\mathcal{L}^{\alpha=1,3}(S^1)$ carry the half-integral monopole charge. Note also that, since the chiral multiplet $\Phi_1$ has the $U(1)$ charge $Q_1 = 2$, the half-integral flux $m_1$ also obeys the Dirac quantization condition properly.

We propose that the above half-indices (38) and (42) in the degenerate limit $\nu \to -1$ and $\eta \to +1$ agree with the first four characters $\chi_0(q)$, $\chi_1(q)$, $\chi_2(q)$, and $\chi_3(q)$ of the non-unitary Haagerup RCFT

$$\chi_0(q) = q^{\Delta_0} \sum_{\mathfrak{m} \in \mathcal{Z}} \frac{q^{\frac{1}{2}\mathfrak{m}^T \cdot K \cdot \mathfrak{m} + (r-1)m_1 + \sum_{a=1}^{r}(a-1)m_a}}{(q)_{2m_1}(q)_{m_2} \ldots (q)_{m_r}},$$

$$\chi_1(q) = q^{\Delta_1} \sum_{\mathfrak{m} \in \mathcal{Z}'} \frac{q^{\frac{1}{2}\mathfrak{m}^T \cdot K \cdot \mathfrak{m} + (r-1)m_1 + \sum_{a=1}^{r}(a-1)m_a - \frac{3(r-1)}{4}}}{(q)_{2m_1}(q)_{m_2} \ldots (q)_{m_r}},$$

$$\chi_2(q) = q^{\Delta_2} \sum_{\mathfrak{m} \in \mathcal{Z}} \frac{q^{\frac{1}{2}\mathfrak{m}^T \cdot K \cdot \mathfrak{m}}}{(q)_{2m_1}(q)_{m_2} \ldots (q)_{m_r}},\tag{44}$$

$$\chi_3(q) = q^{\Delta_3} \sum_{\mathfrak{m} \in \mathcal{Z}'} \frac{q^{\frac{1}{2}\mathfrak{m}^T \cdot K \cdot \mathfrak{m} - \frac{(r-1)}{4}}}{(q)_{2m_1}(q)_{m_2} \ldots (q)_{m_r}},$$

up to the prefactors $q^\Delta$. The exponents $\Delta_0 = -c/24$ and $\Delta_\alpha = h_\alpha - c/24$, and they account for the ground state energies in the corresponding Hilbert spaces $\mathcal{H}$ and $\mathcal{H}^\alpha$. From the bulk computation of modular data in (19), we can determine them only modulo 1 and they are given by

$$(\Delta_0, \Delta_1, \Delta_2, \Delta_3) = \left(\frac{6k+11}{24}, -\frac{1}{24}, -\frac{1}{24}, \frac{6k+11}{24}\right) \pmod 1. \tag{45}$$

The connection between the half-indices of three-dimensional $\mathcal{N} \geq 2$ gauge theories and the conformal characters of two-dimensional CFTs has been explored in numerous recent studies. For instance, it was shown in [54] that the conformal characters of the $G_k$ WZW models with $k > 0$ agrees with the half-indices of 3D $\mathcal{N} = 2$ pure $G_{k+h}$ Chern-Simons theories where $h$ denotes the dual Coxeter number of $G$. In fact, the 3D SUSY theories flow to the pure bosonic $G_k$ Chern-Simons theories in the infrared limit. The relation between the characters of 2D logarithmic CFTs and the half-indices of 3D $\mathcal{N} = 2$ theories has been studied in [55–57]. More recently, Virasoro characters of the non-unitary minimal models are obtained from the half-indices of 3D $\mathcal{N} = 4$ rank-0 SCFTs [36]. Our proposal is parallel to these ideas.

Armed with the above four characters (44), we obtain in the next Section the explicit $q$-expansion of all conformal characters $\{\chi_\alpha(q)\}$ ($\alpha = 0, 1, .., 2k+1$) of the Haagerup RCFTs $\mathcal{R}_k$ with $3 \leq k \leq 11$ using two well-known methods.

Moreover, given such characters, one can express the conformal characters $\widetilde{\chi}_\alpha(q)$ of the $\mathbb{Z}_2$ orbifold RCFT $\widetilde{\mathcal{R}}_k$, the boundary RCFT for the $\mathbb{Z}_2$ gauged TQFT TFT$_-[\widetilde{\mathcal{S}}_k]$ (28), as follows:

**For $k \in 4\mathbb{Z}$**  ($\alpha = 2, .., k/4$ and $\beta = (k+4)/4, .., k/2$)

$$
\begin{aligned}
\widetilde{\chi}_0(q) &= \chi_0(q) + \chi_1(q) = q^{\Delta_0} \sum_{\mathbf{m} \in \mathcal{Z} \oplus \mathcal{Z}'} \frac{q^{\frac{1}{2}\mathbf{m}^T \cdot K \cdot \mathbf{m} + (r-1)m_1 + \sum_{a=1}^{r}(a-1)m_a}}{(q)_{2m_1}(q)_{m_2}\ldots(q)_{m_r}}, \\
\widetilde{\chi}_1(q) &= \chi_2(q) + \chi_3(q) = q^{\Delta_2} \sum_{\mathbf{m} \in \mathcal{Z} \oplus \mathcal{Z}'} \frac{q^{\frac{1}{2}\mathbf{m}^T \cdot K \cdot \mathbf{m}}}{(q)_{2m_1}(q)_{m_2}\ldots(q)_{m_r}}, \\
\widetilde{\chi}_\alpha(q) &= \chi_{2\alpha+1}(q) + \chi_{k+3-2\alpha}(q), \\
\widetilde{\chi}_\beta(q) &= \chi_{2\beta+\frac{k}{2}}(q) + \chi_{\frac{5k}{2}+2-2\beta}(q).
\end{aligned} \tag{46}
$$

**For $k \in 4\mathbb{Z}+2$**  ($\alpha = 2, .., (k-2)/4$ and $\beta = (k+10)/4, .., (k+2)/2$)

$$
\begin{aligned}
\widetilde{\chi}_0(q) &= \chi_0(q) + \chi_1(q) = q^{\Delta_0} \sum_{\mathbf{m} \in \mathcal{Z} \oplus \mathcal{Z}'} \frac{q^{\frac{1}{2}\mathbf{m}^T \cdot K \cdot \mathbf{m} + (r-1)m_1 + \sum_{a=1}^{r}(a-1)m_a}}{(q)_{2m_1}(q)_{m_2}\ldots(q)_{m_r}}, \\
\widetilde{\chi}_1(q) &= \chi_2(q) + \chi_3(q) = q^{\Delta_2} \sum_{\mathbf{m} \in \mathcal{Z} \oplus \mathcal{Z}'} \frac{q^{\frac{1}{2}\mathbf{m}^T \cdot K \cdot \mathbf{m}}}{(q)_{2m_1}(q)_{m_2}\ldots(q)_{m_r}}, \\
\widetilde{\chi}_\alpha(q) &= \chi_{2\alpha+1}(q) + \chi_{k+3-2\alpha}(q), \\
\widetilde{\chi}_{\frac{k}{4}+\frac{1}{2}}(q) &= \widetilde{\chi}_{\frac{k}{4}+\frac{3}{2}}(q) = \chi_{\frac{k}{2}+2}(q), \\
\widetilde{\chi}_\beta(q) &= \chi_{\frac{k}{2}+2\beta-3}(q) + \chi_{\frac{5k}{2}+5-2\beta}(q), \\
\widetilde{\chi}_{\frac{k}{2}+2}(q) &= \widetilde{\chi}_{\frac{k}{2}+3}(q) = \chi_{\frac{3k}{2}+1}(q).
\end{aligned} \tag{47}
$$

Note that the characters $\tilde{\chi}_0$ and $\tilde{\chi}_1$ above are expressed as a fermionic sum, also known as Nahm sums [58–63] after rescaling $m_1$ to $\frac{1}{2}m_1$. It provide a new infinite series of modular forms in the Nahm sum representation.

**For $k \in 4\mathbb{Z} \pm 1$** $\quad$ ($\alpha = 2, .., (k-1)/2$ and $\beta = (k+1)/2, .., k$)

$$
\begin{aligned}
\widetilde{\chi}_0(q) &= \chi_0(q)\chi_1^{\pm}(q) + \chi_1(q)\chi_0^{\pm}(q), \\
\widetilde{\chi}_1(q) &= \chi_2(q)\chi_0^{\pm}(q) + \chi_3(q)\chi_1^{\pm}(q), \\
\widetilde{\chi}_{\alpha}(q) &= \chi_{k+4-2\alpha}(q)\chi_0^{\pm}(q) + \chi_{2\alpha}(q)\chi_1^{\pm}(q), \\
\widetilde{\chi}_{\beta}(q) &= \chi_{3k+2-2\beta}(q)\chi_0^{\pm}(q) + \chi_{2\beta}(q)\chi_1^{\pm}(q).
\end{aligned}
\tag{48}
$$

Here $\chi_{0,1}^{\pm}(q)$ are the conformal characters of a RCFT corresponding to the bulk theory $\mathcal{A}_{\mathbb{Z}_2}^{\pm}$ in (26).

## 3.2 Modular linear differential equation

A simple and practical approach to construct a vector-valued modular form is the method of holomorphic modular boostrap, which is also known as the modular linear differential equation (MLDE). A general form of $d$-th order MLDE can be expressed as

$$
\left[ \mathcal{D}^d + \sum_{s=0}^{d-1} \phi_s(\tau)\mathcal{D}^s \right] f(\tau) = 0,
\tag{49}
$$

where $\phi_s(\tau)$ are weakly holomorphic modular forms of weight $2(d-s)$. Here the $s$-th order derivative $\mathcal{D}^s$ is defined as

$$
\mathcal{D}^s = \mathcal{D}_{2s-2}\mathcal{D}_{2s-4}\cdots\mathcal{D}_0,
\tag{50}
$$

where $\mathcal{D}_t$ is the Serre derivative

$$
\mathcal{D}_t = \frac{1}{2\pi i}\frac{d}{d\tau} - \frac{t}{12}E_2(\tau),
\tag{51}
$$

that maps a modular form of weight $t$ to a modular form of weight $t + 2$. $E_2(\tau)$ denotes the Eisenstein series of weight two. Since (49) is invariant under the modular transformation, one can argue that its $d$ independent solutions have to transform as a $d$-dimensional vector-valued modular function under $SL(2,\mathbb{Z})$.

There are two useful parameters that provides a organizing principle to explore the space of solutions to MDLEs. One of them is the number of independent solutions $d$, and the other is the so-called Wronskian index $l$ that constrains the number of allowed singularities of the the the coefficients $\phi_k(\tau)$ in the fundamental domain. For a given $l$, one can determine $\phi_k(\tau)$ with finitely many unknown parameters, which makes the MLDE method highly practical. Note that one can also desrcribe the Wronskian index $l$ as

$$
l = \frac{d(d-1)}{2} - 6\sum_{i=0}^{d-1}\Delta_i,
\tag{52}
$$

where $\Delta_i$ are the leading exponents of the solutions in the limit $q \to 0$. For more details, please see [64] and the reference therein.

The question is then how to pin down the MLDEs whose solutions can be identified as the conformal characters of the Haagerup RCFTs. To this end, we demand that the above four conformal characters (44) are solutions to the MLDE. Since the Fourier coefficients of such characters are all specified, the finitely many parameters fixing the coefficients $\phi_p(\tau)$ become rigid unless the value of the Wronskian index $l$ changes.

However the MLDE approach to our problem has a small drawback. There is no physical input to fix the value of the Wronskian index $l$ a priori. The bulk description, either the Haagerup TQFTs or their SCFTs, only provides the values of $\Delta_i$ modulo integer. We thus have to examine the space of possible MLDEs by scanning over different values of $l$ until the solutions give rise to a vvmf that only admits non-negative integer Fourier coefficients. One can then regard this vvmf as the conformal characters associated with the bulk TQFTs.

**Example**   To demonstrate this procedure, let us construct the conformal characters of the orbifold Haagerup RCFT $\widetilde{\mathcal{R}}_k$ with $k = 6$.

Since TFT$_-[\widetilde{\mathcal{S}}_{k=6}]$ is a non-spin TQFT, the putative RCFT of our interest has to be bosonic. It has seven conformal characters $\widetilde{\chi}_\alpha(\tau)$ (47), but only five are independent. This is because the charge conjugation matrix $\widetilde{C} = \widetilde{S}^2$ where $\widetilde{S}$ is the modular S-matrix (30)

$$
\widetilde{S} = \begin{pmatrix}
\frac{\sqrt{2}+1}{4} & \frac{\sqrt{2}-1}{4} & \frac{\sqrt{2}}{4} & \frac{\sqrt{2}}{4} & -\frac{1}{2} & -\frac{1}{4} & -\frac{1}{4} \\
\frac{\sqrt{2}-1}{4} & \frac{\sqrt{2}+1}{4} & \frac{1}{2\sqrt{2}} & \frac{1}{2\sqrt{2}} & \frac{1}{2} & \frac{1}{4} & \frac{1}{4} \\
\frac{1}{2\sqrt{2}} & \frac{1}{2\sqrt{2}} & -\frac{1-i}{2\sqrt{2}} & -\frac{1+i}{2\sqrt{2}} & 0 & \frac{i}{2\sqrt{2}} & -\frac{i}{2\sqrt{2}} \\
\frac{1}{2\sqrt{2}} & \frac{1}{2\sqrt{2}} & -\frac{1+i}{2\sqrt{2}} & -\frac{1-i}{2\sqrt{2}} & 0 & -\frac{i}{2\sqrt{2}} & \frac{i}{2\sqrt{2}} \\
-\frac{1}{2} & \frac{1}{2} & 0 & 0 & 0 & -\frac{1}{2} & -\frac{1}{2} \\
-\frac{1}{4} & \frac{1}{4} & \frac{i}{2\sqrt{2}} & -\frac{i}{2\sqrt{2}} & -\frac{1}{2} & \frac{1-i\sqrt{2}}{4} & \frac{1+i\sqrt{2}}{4} \\
-\frac{1}{4} & \frac{1}{4} & -\frac{i}{2\sqrt{2}} & \frac{i}{2\sqrt{2}} & -\frac{1}{2} & \frac{1+i\sqrt{2}}{4} & \frac{1-i\sqrt{2}}{4}
\end{pmatrix} ,
\tag{53}
$$

exchanges $\widetilde{\chi}_2 \leftrightarrow \widetilde{\chi}_3$ and $\widetilde{\chi}_5 \leftrightarrow \widetilde{\chi}_6$, and thus those characters become degenerate, $\widetilde{\chi}_2(\tau) = \widetilde{\chi}_3(\tau)$ and $\widetilde{\chi}_5(\tau) = \widetilde{\chi}_6(\tau)$.

From (47), one can also read off the explicit $q$-expansion of the first two characters

$$
\begin{aligned}
\widetilde{\chi}_0 &= q^{\widetilde{\Delta}_0} \sum_{\mathfrak{m} \in \mathcal{Z} \oplus \mathcal{Z}'} \frac{q^{\frac{1}{2}\mathfrak{m}^T.K.\mathfrak{m}+4m_1+m_2+2m_3+3m_4+4m_5}}{(q)_{2m_1}(q)_{m_2}(q)_{m_3}(q)_{m_4}(q)_{m_5}} \\
&= q^{\widetilde{\Delta}_0}\left(1 + q^2 + 2q^3 + 3q^4 + 4q^5 + 7q^6 + 8q^7 + 14q^8 + 18q^9 + \ldots\right) ,
\end{aligned}
\tag{54}
$$

and

$$
\begin{aligned}
\widetilde{\chi}_1 &= q^{\widetilde{\Delta}_1} \sum_{\mathfrak{m} \in \mathcal{Z} \oplus \mathcal{Z}'} \frac{q^{\frac{1}{2}\mathfrak{m}^T.K.\mathfrak{m}}}{(q)_{2m_1}(q)_{m_2}(q)_{m_3}(q)_{m_4}(q)_{m_5}} \\
&= q^{\widetilde{\Delta}_1}\left(1 + 2q + 4q^2 + 6q^3 + 11q^4 + 16q^5 + 25q^6 + 36q^7 + 54q^8 + 76q^9 + \ldots\right) ,
\end{aligned}
\tag{55}
$$

where the $K$ matrix (4) is

$$
K = \begin{pmatrix}
8 & 0 & 2 & 4 & 6 \\
0 & 2 & 2 & 2 & 2 \\
2 & 2 & 4 & 4 & 4 \\
4 & 2 & 4 & 6 & 6 \\
6 & 2 & 4 & 6 & 8
\end{pmatrix} .
\tag{56}
$$

Here the exponents are $\widetilde{\Delta}_0 = \Delta_0$ and $\widetilde{\Delta}_1 = \Delta_2$. Note that the descendants of each character only carry integral conformal weights relative to that of the primary. This is consistent with the fact that the orbifold with $k = 6$ is bosonic.

To construct the rests of the conformal characters, we have to determine an MLDE. Since there are five independent conformal characters, the order of the MLDE is fixed by five, $d = 5$. Let us start the exploration with the lowest value of the Wronskian index $l = 0$, and see if the solutions have non-negative integer Fourier coefficients. The most general MLDE with $d = 5$ and $l = 0$ can be described as

$$
\left[\mathcal{D}^5 + \mu_1 E_4(\tau)\mathcal{D}^3 + \mu_2 E_6(\tau)\mathcal{D}^2 + \mu_3 E_4^2(\tau)\mathcal{D} + \mu_4 E_4(\tau)E_6(\tau)\right]f(\tau) = 0,
\tag{57}
$$

where $E_4(\tau)$ and $E_6(\tau)$ are the Eisenstein series of weight four and six. We then require that either (54) or (55) is a solution to the MLDE (57), which fixes the parameters uniquely as

$$
\mu_1 = -\frac{24}{144}, \qquad \mu_2 = \frac{65}{2304}, \qquad \mu_3 = -\frac{965}{331776}\nu, \qquad \mu_4 = \frac{1265}{3981312}.
\tag{58}
$$

Having fixed the MLDE completely, we solve (57) using the Frobenius method to obtain the candidate conformal characters in powers of $q$,

$$\widetilde{\chi}_0(q) = q^{\frac{23}{24}}\left(1 + q^2 + 2q^3 + 3q^4 + 4q^5 + 7q^6 + 8q^7 + 14q^8 + 18q^9 + \dots\right),$$

$$\widetilde{\chi}_1(q) = q^{-\frac{1}{24}}\left(1 + 2q + 4q^2 + 6q^3 + 11q^4 + 16q^5 + 25q^6 + 36q^7 + 54q^8 + 76q^9 + \dots\right),$$

$$\widetilde{\chi}_2(q) = \widetilde{\chi}_3(q) = q^{\frac{5}{24}}\left(1 + q + 3q^2 + 4q^3 + 7q^4 + 10q^5 + 17q^6 + 23q^7 + 35q^8 + \dots\right),$$

$$\widetilde{\chi}_4(q) = q^{\frac{1}{12}}\left(1 + 2q + 3q^2 + 6q^3 + 9q^4 + 14q^5 + 22q^6 + 32q^7 + 46q^8 + \dots\right),$$

$$\widetilde{\chi}_5(q) = \widetilde{\chi}_6(q) = q^{\frac{11}{24}}\left(1 + q + 2q^2 + 3q^3 + 6q^4 + 8q^5 + 13q^6 + 18q^7 + 27q^8 + \dots\right).$$

We can see that all exponents are rational-valued and agree with (30), and that Fourier coefficients are non-negative integer-valued. One can also show numerically that, under the modular transformation $\tau \to -1/\tau$,

$$\widetilde{\chi}_\alpha(-1/\tau) = \sum_{\beta=0}^{6}\widetilde{S}_{\alpha\beta}\,\widetilde{\chi}_\beta(\tau), \tag{59}$$

with the modular S-matrix $\widetilde{S}$ (53). In the numerical check, we choose the value of $\tau$ near $i$ so that both $q := e^{2\pi i\tau}$ and $\tilde{q} := e^{2\pi i(-1/\tau)}$ are small. Thus, we can conclude that (**??**) are the very conformal characters of the orbifold.

Several remarks are in order. Relying on our experiences with the lower values of $k$, we propose that the candidate characters of the orbifold $\widetilde{\mathcal{R}}_k$ for $k \in 4\mathbb{Z}_{\geq 1} + 2$ have the leading exponents as follows,

$$\widetilde{\Delta}_\alpha = \left(\frac{k-2}{4},\ 0,\ \underbrace{\frac{n^2}{k-2}}_{n=1,2,..,(k-6)/4},\ \frac{k-2}{16},\ \frac{k-2}{16},\ \underbrace{\frac{m^2}{k+2}}_{m=1,2,..,(k-2)/4},\ \frac{k+2}{16},\ \frac{k+2}{16}\right) - \frac{1}{24}. \tag{60}$$

It implies that, as $k$ increases, the values of $d$ and $l$ grow rapidly, making the MLDE approach less practical. In addition, the MLDE technique applied to the Haagerup RCFTs shares the same challenges with their $\mathbb{Z}_2$ orbifolds. Notice that the $c_{\text{eff}} := -24 \times \min_\alpha\{\tilde{\Delta}_\alpha\}$ is always 1, which is expected to be also true for the $\mathcal{R}_k$s.

To address this challeges, we shall employ an alternative method to construct a vvmf based on a given modular data and four characters (44) in what follows.

## 3.3 Riemann-Hilbert method

It was shown in [8,65–67] that there exists a characteristic matrix $\Xi(\tau)$ that describes the space of vvmfs for a given modular data such as $S$ and $T$ matrices. More precisely, each column of $\Xi(\tau)$ becomes a generator that transforms as a given $d$-dimensional representation of $SL(2,\mathbb{Z})$. Any such vvmf $\{\xi_\alpha(\tau)\}$ can thus be expressed as

$$\xi_\alpha(\tau) = \Xi_{\alpha\beta}(\tau)P_\beta\big(j(\tau)\big), \tag{61}$$

where $P_\beta\big(j(\tau)\big)$ are polynomials of the Klein $j$-function,

$$j(\tau) = \frac{1728E_4^3(\tau)}{E_4^3(\tau) - E_6^2(\tau)}. \tag{62}$$

This approach is known as the theory of Bantay and Gannon [8], which is also referred to as the Riemann-Hilbert method in the literature.

The analytic construction of the matrix $\Xi(\tau)$ needs two constant matrices, denoted by $\Lambda$ and $\chi$. We first briefly discuss the constraints on the pair of two matrices. $\Lambda$ is a diagonal matrix with

$$\exp\left[2\pi i\Lambda\right] = T\,, \tag{63}$$

and satisfies the relation below

$$\text{Tr}\,\Lambda = -\frac{7d}{12} + \frac{1}{4}\text{Tr}\,S + \frac{2}{3\sqrt{3}}\text{Re}\left[e^{-\frac{\pi i}{6}}\text{Tr}\left(ST^{-1}\right)\right]\,. \tag{64}$$

Here $S$ and $T$ are given $d\times d$ modular S- and T-matrices. We define two $d\times d$ matrices $B_2$ and $B_3$,

$$\begin{aligned}
B_2 &= -\frac{31}{72}\Lambda - \frac{1}{1728}\left(\chi + [\Lambda,\chi]\right)\,, \\
B_3 &= -\frac{41}{72}\Lambda + \frac{1}{1728}\left(\chi + [\Lambda,\chi]\right)\,,
\end{aligned} \tag{65}$$

to describe the conditions that the other constant matrix $\chi$ should obey,

$$B_2\left(B_2 - \frac{1}{2}\right) = 0\,, \quad B_3\left(B_3 - \frac{1}{3}\right)\left(B_3 - \frac{2}{3}\right) = 0\,. \tag{66}$$

Given two constant matrices $\Lambda$ and $\chi$, let a $d\times d$ matrix $\Xi(\tau)$ be expanded in powers of $q$ as

$$\Xi(\tau) = q^\Lambda \sum_{n=0}^\infty \Xi[n]q^n\,, \tag{67}$$

with $\Xi[0] = \mathbf{1}_d$ and $\Xi[1] = \chi$.

One can show that any vvmf transforming as a given representation of $SL(2,\mathbb{Z})$ can be generated by the columns of $\Xi(\tau)$, when the other matrices $\Xi[n]$ in (67) satisfy the recursion relation below,

$$n\Xi[n] + \left[\Lambda, \Xi[n]\right] = \sum_{i=0}^{n-1}\Xi[i]\left\{f_{n-i}\Lambda + g_{n-i}\left(\chi + [\Lambda,\chi]\right)\right\}\,, \tag{68}$$

where $f_i$ and $g_i$ are defined as

$$\begin{aligned}
\left(j(\tau) - 984\right)\frac{\Delta(\tau)}{E_{10}(\tau)} &= \sum_{n=0}^\infty f_n q^n\,, \\
\frac{\Delta(\tau)}{E_{10}(\tau)} &= \sum_{n=0}^\infty g_n q^n\,.
\end{aligned} \tag{69}$$

Here $\Delta(\tau)$ is the modular discriminant,

$$\Delta(\tau) = \frac{E_4^3(\tau) - E_6^2(\tau)}{1728}\,. \tag{70}$$

For the sake of self-containedness, let us briefly sketch the key ideas of the Bantay-Gannon method. When a given modular representation admits a diagonal $T$ matrix, it was proven that there always exists the bijective exponent $\Lambda$ that must obey the trace formula (64). By bijective exponent, we mean that any vvmf $\chi_\alpha(\tau)$ for the given modular representation can be expressed in powers of $q$ as

$$\chi_\alpha(\tau) = q^{\Lambda_{ii}}\sum_{\substack{n\in\mathbb{Z}\\n\gg-\infty}} b_\alpha(n)q^n\,, \tag{71}$$

where the principal part $b_\alpha(n)$ with $n \le 0$ uniquely determines the rest of the coefficients $b_\alpha(n)$ with $n > 0$. Suppose that each column of $\Xi(\tau)$ defined in (67) transforms as a vvmf for the given modular matrices. One can then argue that

$$\frac{E_4(\tau)E_6(\tau)}{\Delta(\tau)}\mathcal{D}\Xi(\tau) = \Xi(\tau)\mathfrak{P}(j(\tau)), \tag{72}$$

where the polynomial matrix $\mathfrak{P}$ is

$$\mathfrak{P}(j(\tau)) = (j(\tau) - 984)\Lambda + \chi + [\Lambda, \chi]. \tag{73}$$

Notice that the both sides of (72) are vvmfs that share the same principal part. The bijective exponent $\Lambda$ guarantees that they should be equal. The recursion relation (68) follows directly from (72). Since each column of $\Xi(\tau)$ has a simple principal part, one can easily read off the polynomials of (61) such that the principal part of $\xi_\alpha(\tau)$ matches that of a character $\chi_\alpha(\tau)$ of our interest. Again, the bijective $\Lambda$ implies that $\chi_\alpha(\tau)$ has to agree with $\xi_\alpha(\tau)$.

For a given modular data, it is rather non-trivial to determine the two constant matrices $\Lambda$ and $\chi$ solving (66) in general. As we will see shortly, the known explicit $q$-expansions of four characters (44) plays a crucial role in further constraining the matrix $\chi$, and make the Riemann-Hilbert method higly effective. It eventually leads us to determine the characteristic matrix $\Xi(\tau)$ completely. Once $\Xi(\tau)$ is obtained, it is straightforward to construct the conformal characters of the Haagerup RCFTs.

**The Haagerup RCFT $\mathcal{R}_{k=4}$**  To illustrate, let us describe the procedure for determining the conformal characters of the Haagerup RCFT $\mathcal{R}_k$ with $k = 4$ via the Riemann-Hilbert method.

Given the modular matrices (19), the conformal characters $\{\chi_\alpha(\tau)\}$ of $\mathcal{R}_{k=4}$ transform as a ten-dimensional representation of $SL(2,\mathbb{Z})$. Thus, one can describe them in terms of a $10 \times 10$ characteristic matrix $\Xi(\tau)$. To generate $\Xi(\tau)$, it is essential to construct two constant matrices $\Lambda$ and $\chi$. We elaborate on how to determine them in what follows.

The constant matrix $\Lambda$ should be compatible with the modular T-matrix (63) and its trace is constrained to be equal to $-5$ due to (64). Our choice of $\Lambda$ is

$$\Lambda = \mathrm{diag}\left(-\frac{13}{24}, -\frac{1}{24}, -\frac{1}{24}, -\frac{13}{24}, -\frac{11}{12}, -1, -\frac{7}{8}, -\frac{2}{3}, -\frac{3}{8}, 0\right). \tag{74}$$

Each eigenvalue of $\Lambda$ is equal to the leading exponent $\Delta_\alpha$ of each character $\chi_\alpha(\tau)$ modulo integer. In fact, solving the modular crossing equation numerically,

$$\chi_\alpha(-1/\tau) = \sum_\beta S_{\alpha\beta}\chi_\beta(\tau), \tag{75}$$

one can read off the exponent $\Delta_\alpha$ of each character. In particular, the exponents of the first four characters are given by

$$\Delta_0 = \frac{11}{24}, \qquad \Delta_1 = \frac{47}{24}, \qquad \Delta_2 = -\frac{1}{24}, \qquad \Delta_3 = \frac{11}{24}. \tag{76}$$

The next is to determine another constant matrix $\chi$ that solves the conditions (66). To this end, we first note that there exists two pairs of conformal characters, $(\chi_0(\tau), \chi_3(\tau))$ or $(\chi_1(\tau), \chi_2(\tau))$, whose conformal weights differ by an integer. For each pair, any linear combination of two characters remains an eigenvector of the modular T-matrix $T$. One can also see that the combinations

$$\chi_0(\tau) + \chi_3(\tau), \qquad \chi_1(\tau) + \chi_2(\tau), \tag{77}$$

together with $\chi_4(\tau)$ transform among themselves and decouple from the rests under the S transformation. This implies that the modular S-matrix can be block-diagonalized as follows

$$S = U^{-1} \begin{pmatrix} S_1 & \\ & S_2 \end{pmatrix} U, \tag{78}$$

where each block becomes a congruence representation of $SL(2, \mathbb{Z})$,

$$S_1 = \begin{pmatrix} \frac{1}{2} & \frac{1}{2} & -\frac{1}{\sqrt{2}} \\ \frac{1}{2} & \frac{1}{2} & \frac{1}{\sqrt{2}} \\ -\frac{1}{\sqrt{2}} & \frac{1}{\sqrt{2}} & 0 \end{pmatrix}, \tag{79}$$

and

$$S_2 = \begin{pmatrix} \frac{1}{2\sqrt{3}} & \frac{1}{2\sqrt{3}} & -\frac{1}{\sqrt{6}} & -\frac{1}{\sqrt{6}} & -\frac{1}{\sqrt{6}} & -\frac{1}{\sqrt{6}} & -\frac{1}{\sqrt{6}} \\ \frac{1}{2\sqrt{3}} & \frac{1}{2\sqrt{3}} & \frac{1}{\sqrt{6}} & -\frac{1}{\sqrt{6}} & \frac{1}{\sqrt{6}} & \frac{1}{\sqrt{6}} & \frac{1}{\sqrt{6}} \\ -\frac{1}{\sqrt{6}} & \frac{1}{\sqrt{6}} & \frac{1}{2} & \frac{1}{2\sqrt{3}} & 0 & -\frac{1}{2\sqrt{3}} & -\frac{1}{2} \\ -\frac{1}{\sqrt{6}} & -\frac{1}{\sqrt{6}} & \frac{1}{2\sqrt{3}} & -\frac{1}{2\sqrt{3}} & -\frac{1}{\sqrt{3}} & \frac{1}{2\sqrt{3}} & \frac{1}{2\sqrt{3}} \\ -\frac{1}{\sqrt{6}} & \frac{1}{\sqrt{6}} & 0 & -\frac{1}{\sqrt{3}} & 0 & \frac{1}{\sqrt{3}} & 0 \\ -\frac{1}{\sqrt{6}} & -\frac{1}{\sqrt{6}} & -\frac{1}{2\sqrt{3}} & \frac{1}{2\sqrt{3}} & \frac{1}{\sqrt{3}} & -\frac{1}{2\sqrt{3}} & -\frac{1}{2\sqrt{3}} \\ -\frac{1}{\sqrt{6}} & \frac{1}{\sqrt{6}} & -\frac{1}{2} & \frac{1}{2\sqrt{3}} & 0 & -\frac{1}{2\sqrt{3}} & \frac{1}{2} \end{pmatrix}. \tag{80}$$

Hence, $\{\chi_\alpha(\tau)\}$ of our interest should be in a reducible representation rather than an irreducible representation. Accordingly, the constant matrix $\chi$ can be described as

$$\chi = U^{-1} \begin{pmatrix} \chi_1 & \\ & \chi_2 \end{pmatrix} U, \tag{81}$$

where $\chi_1$ and $\chi_2$ are $3 \times 3$ and $7 \times 7$ matrices. One can express $\chi$ explicitly as

$$\chi = \begin{pmatrix} \chi_{1,1} & \chi_{1,2} & \chi_{1,3} & \chi_{4,1} & \chi_{1,5} & \chi_{1,6} & \chi_{1,7} & \chi_{1,8} & \chi_{1,9} & \chi_{1,10} \\ \chi_{2,1} & \chi_{2,2} & \chi_{3,2} & \chi_{3,1} & \chi_{2,5} & \chi_{2,6} & \chi_{2,7} & \chi_{2,8} & \chi_{2,9} & \chi_{2,10} \\ \chi_{3,1} & \chi_{3,2} & \chi_{2,2} & \chi_{2,1} & \chi_{2,5} & -\chi_{2,6} & -\chi_{2,7} & -\chi_{2,8} & -\chi_{2,9} & -\chi_{2,10} \\ \chi_{4,1} & \chi_{1,3} & \chi_{1,2} & \chi_{1,1} & \chi_{1,5} & -\chi_{1,6} & -\chi_{1,7} & -\chi_{1,8} & -\chi_{1,9} & -\chi_{1,10} \\ \chi_{5,1} & \chi_{5,2} & \chi_{5,2} & \chi_{5,1} & \chi_{5,5} & 0 & 0 & 0 & 0 & 0 \\ \chi_{6,1} & \chi_{6,2} & -\chi_{6,2} & -\chi_{6,1} & 0 & \chi_{6,6} & \chi_{6,7} & \chi_{6,8} & \chi_{6,9} & \chi_{6,10} \\ \chi_{7,1} & \chi_{7,2} & -\chi_{7,2} & -\chi_{7,1} & 0 & \chi_{7,6} & \chi_{7,7} & \chi_{7,8} & \chi_{7,9} & \chi_{7,10} \\ \chi_{8,1} & \chi_{8,2} & -\chi_{8,2} & -\chi_{8,1} & 0 & \chi_{8,6} & \chi_{8,7} & \chi_{8,8} & \chi_{8,9} & \chi_{8,10} \\ \chi_{9,1} & \chi_{9,2} & -\chi_{9,2} & -\chi_{9,1} & 0 & \chi_{9,6} & \chi_{9,7} & \chi_{9,8} & \chi_{9,9} & \chi_{9,10} \\ \chi_{10,1} & \chi_{10,2} & -\chi_{10,2} & -\chi_{10,1} & 0 & \chi_{10,6} & \chi_{10,7} & \chi_{10,8} & \chi_{10,9} & \chi_{10,10} \end{pmatrix}. \tag{82}$$

For the sake of later convenience, we demand that the conformal characters $\{\chi_\alpha(\tau)\}$ is identified as the the third column of $\Xi(\tau)$. Based on the parameterization (82) with the exponents (76) estimated numerically, this requirement fixes the four unknown parameters by the four characters (44),

$$\chi_{1,3} = 1, \qquad \chi_{3,2} = 0, \qquad \chi_{2,2} = 1, \qquad \chi_{1,2} = 1. \tag{83}$$

Given (74) and (82), one can express the solution to (66) as

$$\chi = \begin{pmatrix}
221 & 1 & 1 & 52 & \chi_{1,5} & -\frac{32x}{\chi_{6,2}} & \chi_{1,7} & \chi_{1,8} & \chi_{1,9} & 0 \\
52\left(25+\frac{756}{x}\right) & 1 & 0 & 52\left(25-\frac{756}{x}\right) & -22\chi_{1,5} & \frac{44928}{\chi_{6,2}} & \frac{5616\chi_{1,7}}{5x} & -\frac{3564\chi_{1,8}}{5x} & \frac{297}{\chi_{9,2}} & 0 \\
52\left(25-\frac{756}{x}\right) & 0 & 1 & 52\left(25+\frac{756}{x}\right) & -22\chi_{1,5} & -\frac{44928}{\chi_{6,2}} & -\frac{5616\chi_{1,7}}{5x} & \frac{3564\chi_{1,8}}{5x} & -\frac{297}{\chi_{9,2}} & 0 \\
52 & 1 & 1 & 221 & \chi_{1,5} & \frac{32x}{\chi_{6,2}} & -\chi_{1,7} & -\chi_{1,8} & -\chi_{1,9} & 0 \\
\frac{53248}{\chi_{1,5}} & -\frac{2048}{\chi_{1,5}} & \frac{2048}{\chi_{1,5}} & \frac{53248}{\chi_{1,5}} & -22 & 0 & 0 & 0 & 0 & 0 \\
-\frac{756\chi_{6,2}}{x} & \chi_{6,2} & -\chi_{6,2} & \frac{756\chi_{6,2}}{x} & 0 & -12 & \frac{24\chi_{1,7}\chi_{6,2}}{35x} & \frac{27\chi_{1,8}\chi_{6,2}}{16x} & -\frac{10\chi_{6,2}}{\chi_{9,2}} & \chi_{6,10} \\
\frac{12285}{\chi_{1,7}} & \frac{35x}{2\chi_{1,7}} & -\frac{35x}{2\chi_{1,7}} & -\frac{12285}{\chi_{1,7}} & 0 & \frac{1120x}{\chi_{1,7}\chi_{6,2}} & -63 & \frac{189\chi_{1,8}}{\chi_{1,7}} & -\frac{210\chi_{1,9}}{\chi_{1,7}} & 0 \\
\frac{24960}{\chi_{1,8}} & -\frac{160x}{27\chi_{1,8}} & \frac{160x}{27\chi_{1,8}} & -\frac{24960}{\chi_{1,8}} & 0 & \frac{160x}{\chi_{1,8}\chi_{6,2}} & \frac{256\chi_{1,7}}{\chi_{1,8}} & 4 & -\frac{320\chi_{1,9}}{\chi_{1,8}} & 0 \\
\frac{19656}{\chi_{1,9}} & \chi_{9,2} & -\chi_{9,2} & -\frac{19656}{\chi_{1,9}} & 0 & -\frac{5760\chi_{9,2}}{\chi_{6,2}} & \frac{832\chi_{1,7}}{5x} & -\frac{1404\chi_{1,8}}{5x} & -87 & 0 \\
-\frac{94068\chi_{10,2}}{x} & \chi_{10,2} & -\chi_{10,2} & \frac{94068\chi_{10,2}}{x} & 0 & -\frac{98370\chi_{10,2}}{\chi_{6,2}} & \frac{6952\chi_{1,7}\chi_{10,2}}{5x} & -\frac{297\chi_{1,8}\chi_{10,2}}{40x} & -\frac{330\chi_{10,2}}{\chi_{9,2}} & 0
\end{pmatrix},$$

where $x = \chi_{1,9}\chi_{9,2}$. One can then compute the characteristic matrix $\Xi(\tau)$ using the recursion relation (68), which leads to the conformal characters

$$\chi_0(\tau) = q^{\frac{11}{24}}\left(1 + \frac{54-x}{54}q + \frac{108-x}{54}q^2 + \cdots\right),$$
$$\chi_1(\tau) = q^{\frac{47}{24}}\left(1 + q + 2q^2 + 3q^3 + \cdots\right),$$
$$\chi_2(\tau) = q^{-\frac{1}{24}}\left(1 + q + 3q^2 + 4q^3 + 7q^4 + \cdots\right),$$
$$\chi_3(\tau) = q^{\frac{11}{24}}\left(1 + \frac{54+x}{54}q + \frac{108+x}{54}q^2 + \cdots\right),$$
$$\chi_4(\tau) = -\frac{2048}{\chi_{1,5}}q^{\frac{1}{12}}\left(1 + 2q + 3q^2 + 6q^3 + 9q^4 + \cdots\right),$$
$$\chi_5(\tau) = -\chi_{6,2}\left(1 + 2q + 4q^2 + 6q^3 + \cdots\right) - \chi_{6,10}\chi_{10,2}\left(q + 2q^2 + 3q^3 + 5q^4 + \cdots\right),$$
$$\chi_6(\tau) = -\frac{35x}{2\chi_{1,7}}q^{\frac{1}{8}}\left(1 + q + 2q^2 + 3q^3 + 6q^4 + \cdots\right),$$
$$\chi_7(\tau) = \frac{160x}{27\chi_{1,8}}q^{\frac{1}{3}}\left(1 + q + 2q^2 + 4q^3 + 6q^4 + \cdots\right),$$
$$\chi_8(\tau) = -\chi_{9,2}q^{\frac{5}{8}}\left(1 + q + 3q^2 + 4q^3 + 7q^4 + \cdots\right),$$
$$\chi_9(\tau) = -\chi_{10,2}q\left(1 + 2q + 3q^2 + 5q^3 + \cdots\right).$$
$$(84)$$

Since the first four characters should agree with (44), one can easily fix a parameter $x$ as $x = 54$. Demanding all $q$-expansion coefficients are non-negative integers, the left-over parameters in $\chi$ can be fixed completely by solving the modular crossing equation (75) numerically. To be concrete, the constant matrix $\chi$ is

$$\chi = \begin{pmatrix}
221 & 1 & 1 & 52 & -2048 & 1728 & -945 & 320 & -54 & 0 \\
2028 & 1 & 0 & 572 & 45056 & -44928 & -19656 & -4224 & -297 & 0 \\
572 & 0 & 1 & 2028 & 45056 & 44928 & 19656 & 4224 & 297 & 0 \\
52 & 1 & 1 & 221 & -2048 & -1728 & 945 & -320 & 54 & 0 \\
-26 & 1 & 1 & -26 & -22 & 0 & 0 & 0 & 0 & 0 \\
14 & -1 & 1 & -14 & 0 & -12 & 12 & -10 & -10 & -1 \\
-13 & -1 & 1 & 13 & 0 & 64 & -63 & -64 & -12 & 0 \\
78 & -1 & 1 & -78 & 0 & -27 & -756 & 4 & 54 & 0 \\
-364 & -1 & 1 & 364 & 0 & -5760 & -2912 & 1664 & -87 & 0 \\
1742 & -1 & 1 & -1742 & 0 & -98370 & 24332 & 44 & -330 & 0
\end{pmatrix}, \quad (85)$$

and thus the conformal characters of the Haagerup RCFT $\mathcal{R}_{k=4}$ are

$$
\begin{aligned}
\chi_0(\tau) &= q^{\frac{11}{24}}\left(1 + q^2 + q^3 + 3q^4 + 3q^5 + 6q^6 + \cdots\right), \\
\chi_1(\tau) &= q^{\frac{47}{24}}\left(1 + q + 2q^2 + 3q^3 + 4q^4 + 6q^5 + 10q^6 + \cdots\right), \\
\chi_2(\tau) &= q^{-\frac{1}{24}}\left(1 + q + 3q^2 + 4q^3 + 7q^4 + 10q^5 + 17q^6 + \cdots\right), \\
\chi_3(\tau) &= q^{\frac{11}{24}}\left(1 + 2q + 3q^2 + 5q^3 + 9q^4 + 13q^5 + 20q^6 + \cdots\right), \\
\chi_4(\tau) &= q^{\frac{1}{12}}\left(1 + 2q + 3q^2 + 6q^3 + 9q^4 + 14q^5 + 22q^6 + \cdots\right), \\
\chi_5(\tau) &= q^0\left(1 + q + 2q^2 + 3q^3 + 5q^4 + 8q^5 + 12q^6 + \cdots\right), \\
\chi_6(\tau) &= q^{\frac{1}{8}}\left(1 + q + 2q^2 + 3q^3 + 6q^4 + 8q^5 + 13q^6 + \cdots\right), \\
\chi_7(\tau) &= q^{\frac{1}{3}}\left(1 + q + 2q^2 + 4q^3 + 6q^4 + 9q^5 + 14q^6 + \cdots\right), \\
\chi_8(\tau) &= q^{\frac{5}{8}}\left(1 + q + 3q^2 + 4q^3 + 7q^4 + 10q^5 + 16q^6 + \cdots\right), \\
\chi_9(\tau) &= q\left(1 + 2q + 3q^2 + 5q^3 + 8q^4 + 12q^5 + 18q^6 + \cdots\right).
\end{aligned}
\tag{86}
$$

The characters coincide with characters of bosonoic theory of supersymmetric $\mathcal{N} = 1$ minimal model $SM(2, 12)$:

$$
\begin{aligned}
\chi^{SM(2,12)}_{(1,1)}(q) &= \chi_0(q) + \chi_1(q), \\
\chi^{SM(2,12)}_{(1,3)}(q) &= \chi_6(q) + \chi_8(q), \\
\chi^{SM(2,12)}_{(1,5)}(q) &= \chi_2(q) + \chi_3(q).
\end{aligned}
\tag{87}
$$

NS characters of $\mathcal{N} = 1$ supersymmetric Virasoro minimal model $SM(P,Q)$ are given as ($1 \le r < P, 1 \le s < Q$ with $r - s \in 2\mathbb{Z}$)

$$
\chi^{SM(P,Q)}_{(r,s)} = q^{h-c/24}\frac{(-q^{1/2}; q)_\infty}{(q)_\infty}\sum_{n\in\mathbb{Z}}\left(q^{(n^2 PQ+n(Qr-Ps))/2} - q^{(nP+r)(nQ+s)/2}\right), \quad \text{with}
$$
$$
h = \frac{(Qr-Ps)^2 - (P-Q)^2}{8PQ}, \qquad c = \frac{3}{2}\left(1 - \frac{2(P-Q)^2}{PQ}\right).
\tag{88}
$$

As a final remark, we propose that $\chi_4(\tau)$, $\chi_5(\tau)$, $\chi_7(\tau)$, and $\chi_9(\tau)$ can be summarized as

$$
\begin{aligned}
\chi_4(\tau) &= \frac{1}{2}\sum_{\mathbf{m}\in\mathcal{Z}\oplus\mathcal{Z}'}\frac{q^{\frac{1}{2}\mathbf{m}^T\cdot K\cdot\mathbf{m}+(1,-1,-1)\cdot\mathbf{m}+\frac{1}{12}}}{(q)_{2m_1}(q)_{m_2}(q)_{m_3}}, \\
\chi_5(\tau) &= \sum_{\mathbf{m}\in\mathcal{Z}}\frac{q^{\frac{1}{2}\mathbf{m}^T\cdot K\cdot\mathbf{m}+(1,0,0)\cdot\mathbf{m}}}{(q)_{2m_1}(q)_{m_2}(q)_{m_3}}, \\
\chi_7(\tau) &= \sum_{\mathbf{m}\in\mathcal{Z}\oplus\mathcal{Z}'}\frac{q^{\frac{1}{2}\mathbf{m}^T\cdot K\cdot\mathbf{m}+(1,1,2)\cdot\mathbf{m}+1/3}}{(q)_{2m_1}(q)_{m_2}(q)_{m_3}}, \\
\chi_9(\tau) &= \sum_{\mathbf{m}\in\mathcal{Z}'}\frac{q^{\frac{1}{2}\mathbf{m}^T\cdot K\cdot\mathbf{m}+(1,0,0)\cdot\mathbf{m}}}{(q)_{2m_1}(q)_{m_2}(q)_{m_3}}.
\end{aligned}
\tag{89}
$$

The matrix $K$ is given in (4) with $k = 4$. It is noteworthy here that five among six modular forms studied in [63] with $A = \begin{pmatrix} 4 & 2 & 1 \\ 2 & 2 & 0 \\ 1 & 0 & 1 \end{pmatrix}$ are present as the above conformal characters $\chi_\alpha(\tau)$ with the summation range slightly modified.

To present the results for other values of $k$ concisely, we only provide essential data $\Lambda$, $\chi$ that generate the characteristic matrix $\Xi(\tau)$. One can always read off the conformal characters from the third column of $\Xi(\tau)$ and confirm that their $q$-expansion coefficients are always non-negative integers.

**The Haagerup RCFT** $\mathcal{R}_{k=3}$    The two constant matrices are given by

$$\Lambda = \mathrm{diag}\left\{-\frac{19}{24}, -\frac{1}{24}, -\frac{1}{24}, -\frac{19}{24}, -\frac{119}{120}, -\frac{101}{120}, -\frac{71}{120}, -\frac{29}{120}\right\}, \tag{90}$$

and

$$\chi = \begin{pmatrix} -171 & 1 & 1 & -76 & 210 & -120 & 45 & -10 \\ 19304 & 2 & 1 & 7448 & -46760 & -17030 & -2520 & -65 \\ 7448 & 1 & 2 & 19304 & 46760 & 17030 & 2520 & 65 \\ -76 & 1 & 1 & -171 & -210 & 120 & -45 & 10 \\ 19 & -1 & 1 & -19 & 21 & -6 & -12 & -8 \\ -56 & -1 & 1 & 56 & 56 & -131 & -56 & 1 \\ 494 & -1 & 1 & -494 & -780 & -1014 & 277 & 12 \\ -4256 & -1 & 1 & 4256 & -23200 & 3585 & 672 & -61 \end{pmatrix}. \tag{91}$$

The third column of $\Xi(\tau)$ gives conformal characters of $\mathcal{R}_{k=3}$, which agree perfectly with the conformal characters of a product of two non-unitary minimal models $M(3,5) \otimes M(2,5)$,

$$
\begin{aligned}
\chi_0(\tau) &= q^{\frac{5}{24}} \sum_{m_1 \in \mathbb{Z}_{\geq 0}, m_2 \in \mathbb{Z}_{\geq 0}} \frac{q^{m_1^2 + m_1 + m_2^2 + m_2}}{(q)_{2m_1}(q)_{m_2}}, \\
\chi_1(\tau) &= q^{\frac{23}{24}} \sum_{m_1 \in \mathbb{Z}_{\geq 0}+\frac{1}{2}, m_2 \in \mathbb{Z}_{\geq 0}} \frac{q^{m_1^2 + m_1 + m_2^2 + m_2 - \frac{3}{4}}}{(q)_{2m_1}(q)_{m_2}}, \\
\chi_2(\tau) &= q^{-\frac{1}{24}} \sum_{m_1 \in \mathbb{Z}_{\geq 0}, m_2 \in \mathbb{Z}_{\geq 0}} \frac{q^{m_1^2 + m_2^2}}{(q)_{2m_1}(q)_{m_2}}, \\
\chi_3(\tau) &= q^{\frac{5}{24}} \sum_{m_1 \in \mathbb{Z}_{\geq 0}+\frac{1}{2}, m_2 \in \mathbb{Z}_{\geq 0}} \frac{q^{m_1^2 + m_2^2 - \frac{1}{4}}}{(q)_{2m_1}(q)_{m_2}}, \\
\chi_4(\tau) &= q^{\frac{1}{120}} \sum_{m_1 \in \mathbb{Z}_{\geq 0}, m_2 \in \mathbb{Z}_{\geq 0}} \frac{q^{m_1^2 + m_1 + m_2^2}}{(q)_{2m_1}(q)_{m_2}}, \\
\chi_5(\tau) &= q^{\frac{19}{120}} \sum_{m_1 \in \mathbb{Z}_{\geq 0}, m_2 \in \mathbb{Z}_{\geq 0}} \frac{q^{m_1^2 + m_2^2 + m_2}}{(q)_{2m_1}(q)_{m_2}}, \\
\chi_6(\tau) &= q^{\frac{49}{120}} \sum_{m_1 \in \mathbb{Z}_{\geq 0}+\frac{1}{2}, m_2 \in \mathbb{Z}_{\geq 0}} \frac{q^{m_1^2 + m_2^2 + m_2 - \frac{1}{4}}}{(q)_{2m_1}(q)_{m_2}}, \\
\chi_7(\tau) &= q^{\frac{91}{120}} \sum_{m_1 \in \mathbb{Z}_{\geq 0}+\frac{1}{2}, m_2 \in \mathbb{Z}_{\geq 0}} \frac{q^{m_1^2 + m_1 + m_2^2 - \frac{3}{4}}}{(q)_{2m_1}(q)_{m_2}}.
\end{aligned} \tag{92}
$$

**The Haagerup RCFT $\mathcal{R}_{k=5}$**    The data are given by

$$\Lambda = \mathrm{diag}\left\{-\tfrac{7}{24}, -\tfrac{1}{24}, -\tfrac{1}{24}, -\tfrac{7}{24}, -\tfrac{23}{24}, -\tfrac{17}{24}, -\tfrac{169}{168}, -\tfrac{151}{168}, -\tfrac{121}{168}, -\tfrac{79}{168} -\tfrac{25}{168} -\tfrac{127}{168}\right\},$$

$$\chi = \begin{pmatrix}
-62 & 1 & 1 & -15 & -11178 & 2430 & 9555 & -5152 & 1701 & -280 & 7 & -2184 \\
134 & 1 & 0 & 34 & 48600 & 8262 & -43316 & -21350 & -5950 & -714 & -14 & -7735 \\
34 & 0 & 1 & 134 & 48600 & 8262 & 43316 & 21350 & 5950 & 714 & 14 & 7735 \\
-15 & 1 & 1 & -62 & -11178 & 2430 & -9555 & 5152 & -1701 & 280 & -7 & 2184 \\
-7 & 1 & 1 & -7 & 23 & -34 & 0 & 0 & 0 & 0 & 0 & 0 \\
20 & 1 & 1 & 20 & -552 & -187 & 0 & 0 & 0 & 0 & 0 & 0 \\
5 & -1 & 1 & -5 & 0 & 0 & 13 & -2 & -1 & -10 & -3 & -22 \\
-2 & -1 & 1 & 2 & 0 & 0 & 52 & -27 & -50 & -25 & -2 & 26 \\
12 & -1 & 1 & -12 & 0 & 0 & 156 & -450 & -152 & 46 & 5 & -78 \\
-30 & -1 & 1 & 30 & 0 & 0 & -1300 & -2976 & 798 & 153 & -10 & -338 \\
75 & -1 & 1 & -75 & 0 & 0 & -28900 & -5250 & 6375 & -850 & 6 & 5746 \\
-14 & 0 & 0 & 14 & 0 & 0 & -468 & 225 & -50 & -6 & 6 & -179
\end{pmatrix},$$

and the characters are

$$\chi_0 = q^{\frac{17}{24}}\left(1 + q^2 + q^3 + 2q^4 + 2q^5 + 5q^6 + \cdots\right),$$
$$\chi_1 = q^{\frac{71}{24}}\left(1 + q + 2q^2 + 3q^3 + 4q^4 + 6q^5 + 9q^6 + \cdots\right),$$
$$\chi_2 = q^{-\frac{1}{24}}\left(1 + q + 2q^2 + 4q^3 + 6q^4 + 9q^5 + 14q^6 + \cdots\right),$$
$$\chi_3 = q^{\frac{17}{24}}\left(1 + 2q + 3q^2 + 5q^3 + 8q^4 + 12q^5 + 19q^6 + \cdots\right),$$
$$\chi_4 = q^{\frac{1}{24}}\left(1 + q + 3q^2 + 4q^3 + 8q^4 + 11q^5 + 18q^6 + \cdots\right),$$
$$\chi_5 = q^{\frac{7}{24}}\left(1 + 2q + 3q^2 + 5q^3 + 8q^4 + 13q^5 + 19q^6 + \cdots\right),$$
$$\chi_6 = q^{-\frac{1}{168}}\left(1 + q + 2q^2 + 3q^3 + 5q^4 + 7q^5 + 12q^6 + \cdots\right),$$
$$\chi_7 = q^{\frac{17}{168}}\left(1 + q + 2q^2 + 3q^3 + 5q^4 + 8q^5 + 12q^6 + \cdots\right),$$
$$\chi_8 = q^{\frac{47}{168}}\left(1 + q + 2q^2 + 3q^3 + 6q^4 + 8q^5 + 13q^6 + \cdots\right),$$
$$\chi_9 = q^{\frac{89}{168}}\left(1 + q + 2q^2 + 4q^3 + 6q^4 + 9q^5 + 14q^6 + \cdots\right),$$
$$\chi_{10} = q^{\frac{143}{168}}\left(1 + q + 3q^2 + 4q^3 + 7q^4 + 10q^5 + 16q^6 + \cdots\right),$$
$$\chi_{11} = q^{\frac{209}{168}}\left(1 + 2q + 3q^2 + 5q^3 + 8q^4 + 12q^5 + 18q^6 + \cdots\right).$$

(93)

**The Haagerup RCFT $\mathcal{R}_{k=6}$**    We present two constant matrices below,

$$\Lambda = \mathrm{diag}\left\{-\tfrac{1}{24}, -\tfrac{1}{24}, -\tfrac{1}{24}, -\tfrac{1}{24}, -\tfrac{47}{48}, -\tfrac{19}{24}, -\tfrac{23}{48}, -\tfrac{97}{96}, -\tfrac{11}{12}, -\tfrac{73}{96}, -\tfrac{13}{24}, -\tfrac{25}{96}, -\tfrac{11}{12}, -\tfrac{49}{96},\right\},$$

$$\chi = \begin{pmatrix}
1 & 1 & 1 & 0 & -48128 & 13376 & -1024 & 41600 & -22464 & 7488 & -1300 & 64 & -22592 & 960 \\
1 & 1 & 0 & 1 & 48128 & 13376 & 1024 & -41600 & -22592 & -7488 & -1300 & -64 & -22464 & -960 \\
1 & 0 & 1 & 1 & 48128 & 13376 & 1024 & 41600 & 22592 & 7488 & 1300 & 64 & 22464 & 960 \\
0 & 1 & 1 & 1 & -48128 & 13376 & -1024 & -41600 & 22464 & -7488 & 1300 & -64 & 22592 & -960 \\
-1 & 1 & 1 & -1 & 0 & 0 & -23 & 0 & 0 & 0 & 0 & 0 & 0 & 0 \\
1 & 1 & 1 & 1 & 0 & -247 & 0 & 0 & 0 & 0 & 0 & 0 & 0 & 0 \\
-1 & 1 & 1 & -1 & -4371 & 0 & 253 & 0 & 0 & 0 & 0 & 0 & 0 & 0 \\
1 & -1 & 1 & -1 & 0 & 0 & 0 & -2 & 16 & -14 & 0 & -7 & -16 & -17 \\
0 & -1 & 1 & 0 & 0 & 0 & 0 & 48 & -7 & -40 & -26 & -8 & -15 & 8 \\
1 & -1 & 1 & -1 & 0 & 0 & 0 & 147 & -240 & -182 & 0 & 12 & 240 & -35 \\
-1 & -1 & 1 & 1 & 0 & 0 & 0 & 0 & -2048 & 0 & 273 & 0 & -2048 & 0 \\
1 & -1 & 1 & -1 & 0 & 0 & 0 & -8625 & -9200 & 4575 & 0 & -69 & 9200 & 322 \\
-1 & 0 & 0 & 1 & 0 & 0 & 0 & -48 & -15 & 40 & -26 & 8 & -7 & -8 \\
0 & 0 & 0 & 0 & 0 & 0 & 0 & -3675 & 1792 & -441 & 0 & 21 & -1792 & 256
\end{pmatrix},$$

and characters are

$$\chi_0 = q^{\frac{23}{24}}\left(1 + q^2 + q^3 + 2q^4 + 2q^5 + 4q^6 + \cdots\right),$$
$$\chi_1 = q^{\frac{95}{24}}\left(1 + q + 2q^2 + 3q^3 + 4q^4 + 6q^5 + 9q^6 + \cdots\right),$$
$$\chi_2 = q^{-\frac{1}{24}}\left(1 + q + 2q^2 + 3q^3 + 6q^4 + 8q^5 + 13q^6 + \cdots\right),$$
$$\chi_3 = q^{\frac{23}{24}}\left(1 + 2q + 3q^2 + 5q^3 + 8q^4 + 12q^5 + 18q^6 + \cdots\right),$$
$$\chi_4 = q^{\frac{1}{48}}\left(1 + q + 2q^2 + 4q^3 + 6q^4 + 10q^5 + 15q^6 + \cdots\right),$$
$$\chi_5 = q^{\frac{5}{24}}\left(1 + q + 3q^2 + 4q^3 + 7q^4 + 10q^5 + 17q^6 + \cdots\right),$$
$$\chi_6 = q^{\frac{25}{48}}\left(1 + 2q + 3q^2 + 5q^3 + 8q^4 + 12q^5 + 18q^6 + \cdots\right),$$
$$\chi_7 = q^{-\frac{1}{96}}\left(1 + q + 2q^2 + 3q^3 + 5q^4 + 7q^5 + 11q^6 + \cdots\right), \quad (94)$$
$$\chi_8 = q^{\frac{1}{12}}\left(1 + q + 2q^2 + 3q^3 + 5q^4 + 7q^5 + 12q^6 + \cdots\right),$$
$$\chi_9 = q^{\frac{23}{96}}\left(1 + q + 2q^2 + 3q^3 + 5q^4 + 8q^5 + 12q^6 + \cdots\right),$$
$$\chi_{10} = q^{\frac{11}{24}}\left(1 + q + 2q^2 + 3q^3 + 6q^4 + 8q^5 + 13q^6 + \cdots\right),$$
$$\chi_{11} = q^{\frac{71}{96}}\left(1 + q + 2q^2 + 4q^3 + 6q^4 + 9q^5 + 14q^6 + \cdots\right),$$
$$\chi_{12} = q^{\frac{13}{12}}\left(1 + q + 3q^2 + 4q^3 + 7q^4 + 10q^5 + 16q^6 + \cdots\right),$$
$$\chi_{13} = q^{\frac{143}{96}}\left(1 + 2q + 3q^2 + 5q^3 + 8q^4 + 12q^5 + 18q^6 + \cdots\right).$$

**The Haagerup RCFT $\mathcal{R}_{k=7}$**   Two constant matrices are

$$\Lambda = \mathrm{diag}\left\{-\tfrac{19}{24}, -\tfrac{1}{24}, -\tfrac{1}{24}, -\tfrac{19}{24}, -\tfrac{119}{120}, -\tfrac{101}{120}, -\tfrac{71}{120}, -\tfrac{29}{120}, -\tfrac{73}{72}, -\tfrac{67}{72}, -\tfrac{19}{24}, -\tfrac{43}{72}, -\tfrac{25}{72}, -\tfrac{1}{24}, -\tfrac{49}{72}, -\tfrac{19}{72}\right\},$$

$$\chi = \begin{pmatrix}
-86 & 0 & 0 & -9 & -210 & 120 & -45 & 10 & 162 & -135 & 85 & -36 & 9 & -1 & 45 & -9 \\
10380 & 1 & 0 & 1476 & 46760 & 17030 & 2520 & 65 & -40176 & -23301 & -8924 & -1953 & -180 & -1 & -3816 & -63 \\
1476 & 0 & 1 & 10380 & 46760 & 17030 & 2520 & 65 & 40176 & 23301 & 8924 & 1953 & 180 & 1 & 3816 & 63 \\
-9 & 0 & 0 & -86 & -210 & 120 & -45 & 10 & -162 & 135 & -85 & 36 & -9 & 1 & 45 & 9 \\
-19 & 1 & 1 & -19 & 21 & -6 & -12 & -8 & 0 & 0 & 0 & 0 & 0 & 0 & 0 & 0 \\
56 & 1 & 1 & 56 & 56 & -131 & -56 & 1 & 0 & 0 & 0 & 0 & 0 & 0 & 0 & 0 \\
-494 & 1 & 1 & -494 & -780 & -1014 & 277 & 12 & 0 & 0 & 0 & 0 & 0 & 0 & 0 & 0 \\
4256 & 1 & 1 & 4256 & -23200 & 3585 & 672 & -61 & 0 & 0 & 0 & 0 & 0 & 0 & 0 & 0 \\
7 & -1 & 1 & -7 & 0 & 0 & 0 & 0 & 7 & -5 & 7 & -8 & -5 & -1 & -20 & -7 \\
-20 & -1 & 1 & 20 & 0 & 0 & 0 & 0 & 32 & -2 & -20 & -29 & -12 & -1 & -8 & 1 \\
85 & -1 & 1 & -85 & 0 & 0 & 0 & 0 & 162 & -135 & -162 & -36 & 9 & 1 & 45 & -9 \\
-344 & -1 & 1 & 344 & 0 & 0 & 0 & 0 & 304 & -1279 & -344 & 195 & 40 & -1 & -384 & 2 \\
1735 & -1 & 1 & -1735 & 0 & 0 & 0 & 0 & -2138 & -7285 & 1735 & 764 & -100 & -1 & 1618 & 13 \\
-8924 & -1 & 1 & 8924 & 0 & 0 & 0 & 0 & -40176 & -23301 & 17828 & -1953 & -180 & 2 & -3816 & -63 \\
189 & 0 & 0 & -189 & 0 & 0 & 0 & 0 & -735 & 112 & 189 & -140 & 35 & 0 & -31 & 20 \\
-2808 & 0 & 0 & 2808 & 0 & 0 & 0 & 0 & -19760 & 9984 & -2808 & 260 & 40 & 0 & 2080 & -61
\end{pmatrix},$$

and the characters are

$$\chi_0 = q^{\frac{29}{24}}\left(1 + q^2 + q^3 + 2q^4 + 2q^5 + \cdots\right),$$

$$\chi_1 = q^{\frac{119}{24}}\left(1 + q + 2q^2 + 3q^3 + 4q^4 + 6q^5 + 9q^6 + \cdots\right),$$

$$\chi_2 = q^{-\frac{1}{24}}\left(1 + q + 2q^2 + 3q^3 + 5q^4 + 8q^5 + 12q^6 + \cdots\right),$$

$$\chi_3 = q^{\frac{29}{24}}\left(1 + 2q + 3q^2 + 5q^3 + 8q^4 + 12q^5 + 18q^6 + \cdots\right),$$

$$\chi_4 = q^{\frac{1}{120}}\left(1 + q + 2q^2 + 3q^3 + 6q^4 + 8q^5 + 14q^6 + \cdots\right),$$

$$\chi_5 = q^{\frac{19}{120}}\left(1 + q + 2q^2 + 4q^3 + 6q^4 + 9q^5 + 14q^6 + \cdots\right),$$

$$\chi_6 = q^{\frac{49}{120}}\left(1 + q + 3q^2 + 4q^3 + 7q^4 + 10q^5 + 16q^6 + \cdots\right),$$

$$\chi_7 = q^{\frac{91}{120}}\left(1 + 2q + 3q^2 + 5q^3 + 8q^4 + 12q^5 + 18q^6 + \cdots\right),$$

$$\chi_8 = q^{-\frac{1}{72}}\left(1 + q + 2q^2 + 3q^3 + 5q^4 + 7q^5 + 11q^6 + \cdots\right),$$

$$\chi_9 = q^{\frac{5}{72}}\left(1 + q + 2q^2 + 3q^3 + 5q^4 + 7q^5 + 11q^6 + \cdots\right),$$

$$\chi_{10} = q^{\frac{5}{24}}\left(1 + q + 2q^2 + 3q^3 + 5q^4 + 7q^5 + 12q^6 + \cdots\right),$$

$$\chi_{11} = q^{\frac{29}{72}}\left(1 + q + 2q^2 + 3q^3 + 5q^4 + 8q^5 + 12q^6 + \cdots\right),$$

$$\chi_{12} = q^{\frac{47}{72}}\left(1 + q + 2q^2 + 3q^3 + 6q^4 + 8q^5 + 13q^6 + \cdots\right),$$

$$\chi_{13} = q^{\frac{23}{24}}\left(1 + q + 2q^2 + 4q^3 + 6q^4 + 9q^5 + 14q^6 + \cdots\right),$$

$$\chi_{14} = q^{\frac{95}{72}}\left(1 + q + 3q^2 + 4q^3 + 7q^4 + 10q^5 + 16q^6 + \cdots\right),$$

$$\chi_{15} = q^{\frac{125}{72}}\left(1 + 2q + 3q^2 + 5q^3 + 8q^4 + 12q^5 + 18q^6 + \cdots\right).$$

(95)

**The Haagerup RCFT $\mathcal{R}_{k=8}$**  The seed data $\Lambda$ and $\chi$ are

$$\Lambda = \mathrm{diag}\left\{-\frac{13}{24}, -\frac{1}{24}, -\frac{1}{24}, -\frac{13}{24}, -1, -\frac{7}{8}, -\frac{2}{3}, -\frac{3}{8}, 0, -\frac{61}{60}, -\frac{113}{120}, -\frac{49}{60}, -\frac{77}{120}, -\frac{5}{12}, -\frac{17}{120}, -\frac{49}{60}, -\frac{53}{120}, -\frac{1}{60}\right\},$$

$$\chi = \begin{pmatrix}
148 & 0 & 0 & 21 & -1728 & 945 & -320 & 54 & 0 & 1440 & -1035 & 560 & -210 & 48 & -5 & 560 & -70 & 0 \\
1302 & 1 & 0 & 154 & 44928 & 19656 & 4224 & 297 & 0 & -38720 & -23870 & -10080 & -2695 & -352 & -10 & -10080 & -440 & 0 \\
154 & 0 & 1 & 1302 & 44928 & 19656 & 4224 & 297 & 0 & 38720 & 23870 & 10080 & 2695 & 352 & 10 & 10080 & 440 & 0 \\
21 & 0 & 0 & 148 & -1728 & 945 & -320 & 54 & 0 & -1440 & 1035 & -560 & 210 & -48 & 5 & -560 & 70 & 0 \\
-14 & 1 & 1 & -14 & -12 & 12 & -10 & -10 & -1 & 0 & 0 & 0 & 0 & 0 & 0 & 0 & 0 & 0 \\
13 & 1 & 1 & 13 & 64 & -63 & -64 & -12 & 0 & 0 & 0 & 0 & 0 & 0 & 0 & 0 & 0 & 0 \\
-78 & 1 & 1 & -78 & -27 & -756 & 4 & 54 & 0 & 0 & 0 & 0 & 0 & 0 & 0 & 0 & 0 & 0 \\
364 & 1 & 1 & 364 & -5760 & -2912 & 1664 & -87 & 0 & 0 & 0 & 0 & 0 & 0 & 0 & 0 & 0 & 0 \\
-1742 & 1 & 1 & -1742 & -98370 & 24332 & 44 & -330 & 0 & 0 & 0 & 0 & 0 & 0 & 0 & 0 & 0 & 0 \\
8 & -1 & 1 & -8 & 0 & 0 & 0 & 0 & 0 & -10 & 14 & -7 & 0 & -6 & -2 & -7 & -14 & -1 \\
-7 & -1 & 1 & 7 & 0 & 0 & 0 & 0 & 0 & 32 & 4 & -16 & -22 & -16 & -3 & -16 & -4 & 0 \\
28 & -1 & 1 & -28 & 0 & 0 & 0 & 0 & 0 & 111 & -54 & -133 & -56 & -1 & 2 & 112 & 2 & -1 \\
-62 & -1 & 1 & 62 & 0 & 0 & 0 & 0 & 0 & 384 & -770 & -448 & 77 & 64 & 2 & -448 & -63 & 0 \\
224 & -1 & 1 & -224 & 0 & 0 & 0 & 0 & 0 & -45 & -4970 & 20 & 880 & 2 & -10 & 20 & 210 & 0 \\
-777 & -1 & 1 & 777 & 0 & 0 & 0 & 0 & 0 & -13344 & -21987 & 9744 & 1078 & -496 & 4 & 9744 & -504 & 0 \\
20 & 0 & 0 & -20 & 0 & 0 & 0 & 0 & 0 & -123 & -68 & 119 & -56 & 5 & 4 & -126 & 16 & 1 \\
-210 & 0 & 0 & 210 & 0 & 0 & 0 & 0 & 0 & -4928 & 1254 & 672 & -616 & 160 & -6 & 672 & 73 & 0 \\
1260 & 0 & 0 & -1260 & 0 & 0 & 0 & 0 & 0 & -84843 & 44100 & -13475 & 1848 & 15 & -4 & 13524 & -784 & 1
\end{pmatrix},$$

and characters are

$$
\begin{aligned}
\chi_0 &= q^{\frac{35}{24}}\left(1 + q^2 + q^3 + 2q^4 + 2q^5 + 4q^6 + \cdots\right), \\
\chi_1 &= q^{\frac{143}{24}}(1 + q + 2q^2 + 3q^3 + 4q^4 + 6q^5 + 9q^6 + \cdots) \\
\chi_2 &= q^{-\frac{1}{24}}\left(1 + q + 2q^2 + 3q^3 + 5q^4 + 7q^5 + 12q^6 + \cdots\right), \\
\chi_3 &= q^{\frac{35}{24}}(1 + 2q + 3q^2 + 5q^3 + 8q^4 + 12q^5 + 18q^6 + \cdots), \\
\chi_4 &= 1 + q + 2q^2 + 3q^3 + 5q^4 + 8q^5 + 12q^6 + \cdots, \\
\chi_5 &= q^{\frac{1}{8}}(1 + q + 2q^2 + 3q^3 + 6q^4 + 8q^5 + 13q^6 + \cdots), \\
\chi_6 &= q^{\frac{1}{3}}(1 + q + 2q^2 + 4q^3 + 6q^4 + 9q^5 + 14q^6 + \cdots), \\
\chi_7 &= q^{\frac{5}{8}}(1 + q + 3q^2 + 4q^3 + 7q^4 + 10q^5 + 16q^6 + \cdots), \\
\chi_8 &= q(1 + 2q + 3q^2 + 5q^3 + 8q^4 + 12q^5 + 18q^6 + \cdots), \\
\chi_9 &= q^{-\frac{1}{60}}\left(1 + q + 2q^2 + 3q^3 + 5q^4 + 7q^5 + 11q^6 + \cdots\right), \\
\chi_{10} &= q^{\frac{7}{120}}(1 + q + 2q^2 + 3q^3 + 5q^4 + 7q^5 + 11q^6 + \cdots), \\
\chi_{11} &= q^{\frac{11}{60}}(1 + q + 2q^2 + 3q^3 + 5q^4 + 7q^5 + 11q^6 + \cdots), \\
\chi_{12} &= q^{\frac{43}{120}}(1 + q + 2q^2 + 3q^3 + 5q^4 + 7q^5 + 12q^6 + \cdots), \\
\chi_{13} &= q^{\frac{7}{12}}(1 + q + 2q^2 + 3q^3 + 5q^4 + 8q^5 + 12q^6 + \cdots), \\
\chi_{14} &= q^{\frac{103}{120}}(1 + q + 2q^2 + 3q^3 + 6q^4 + 8q^5 + 13q^6 + \cdots), \\
\chi_{15} &= q^{\frac{71}{60}}(1 + q + 2q^2 + 4q^3 + 6q^4 + 9q^5 + 14q^6 + \cdots), \\
\chi_{16} &= q^{\frac{187}{120}}(1 + q + 3q^2 + 4q^3 + 7q^4 + 10q^5 + 16q^6 + \cdots), \\
\chi_{17} &= q^{\frac{119}{60}}(1 + 2q + 3q^2 + 5q^3 + 8q^4 + 12q^5 + 18q^6 + \cdots).
\end{aligned}
\tag{96}
$$

**The Haagerup RCFT $\mathcal{R}_{k=9}$**  Two constant matrices are presented below,

$$
\Lambda = \mathrm{diag}\left\{-\tfrac{7}{24}, -\tfrac{1}{24}, -\tfrac{1}{24}, -\tfrac{7}{24}, -\tfrac{169}{168}, -\tfrac{151}{168}, -\tfrac{121}{168}, -\tfrac{79}{168}, -\tfrac{25}{168}, -\tfrac{127}{168}, -\tfrac{269}{264}, -\tfrac{251}{264}, -\tfrac{221}{264}, -\tfrac{179}{264}, -\tfrac{125}{264}, -\tfrac{59}{264}, -\tfrac{245}{264}, -\tfrac{155}{264}, -\tfrac{53}{264}, -\tfrac{203}{264}\right\},
$$

$$
\chi = \begin{pmatrix}
-43 & 0 & 0 & -4 & -9555 & 5152 & -1701 & 280 & -7 & 2184 & 8184 & -5610 & 2860 & -1012 & 220 & -22 & 4895 & -550 & 11 & -1892 \\
92 & 1 & 0 & 8 & 43316 & 21350 & 5950 & 714 & 14 & 7735 & -37532 & -24101 & -11154 & -3410 & -594 & -33 & -20658 & -1617 & -22 & -6798 \\
8 & 0 & 1 & 92 & 43316 & 21350 & 5950 & 714 & 14 & 7735 & 37532 & 24101 & 11154 & 3410 & 594 & 33 & 20658 & 1617 & 22 & 6798 \\
-4 & 0 & 0 & -43 & -9555 & 5152 & -1701 & 280 & -7 & 2184 & -8184 & 5610 & -2860 & 1012 & -220 & 22 & -4895 & 550 & -11 & 1892 \\
-5 & 1 & 1 & -5 & 13 & -2 & -1 & -10 & -3 & -22 & 0 & 0 & 0 & 0 & 0 & 0 & 0 & 0 & 0 & 0 \\
2 & 1 & 1 & 2 & 52 & -27 & -50 & -25 & -2 & 26 & 0 & 0 & 0 & 0 & 0 & 0 & 0 & 0 & 0 & 0 \\
-12 & 1 & 1 & -12 & 156 & -450 & -152 & 46 & 5 & -78 & 0 & 0 & 0 & 0 & 0 & 0 & 0 & 0 & 0 & 0 \\
30 & 1 & 1 & 30 & -1300 & -2976 & 798 & 153 & -10 & -338 & 0 & 0 & 0 & 0 & 0 & 0 & 0 & 0 & 0 & 0 \\
-75 & 1 & 1 & -75 & -28900 & -5250 & 6375 & -850 & 6 & 5746 & 0 & 0 & 0 & 0 & 0 & 0 & 0 & 0 & 0 & 0 \\
14 & 0 & 0 & 14 & -468 & 225 & -50 & -6 & 6 & -179 & 0 & 0 & 0 & 0 & 0 & 0 & 0 & 0 & 0 & 0 \\
3 & -1 & 1 & -3 & 0 & 0 & 0 & 0 & 0 & 0 & 11 & 2 & 0 & -4 & -2 & -4 & -17 & -10 & -5 & -18 \\
-2 & -1 & 1 & 2 & 0 & 0 & 0 & 0 & 0 & 0 & 28 & 12 & -14 & -19 & -14 & -7 & -38 & -7 & -2 & 14 \\
7 & -1 & 1 & -7 & 0 & 0 & 0 & 0 & 0 & 0 & 114 & -36 & -91 & -64 & -14 & 2 & 127 & 28 & -1 & -62 \\
-8 & -1 & 1 & 8 & 0 & 0 & 0 & 0 & 0 & 0 & 352 & -457 & -416 & -33 & 64 & 9 & -104 & -133 & -8 & 125 \\
23 & -1 & 1 & -23 & 0 & 0 & 0 & 0 & 0 & 0 & 658 & -3220 & -780 & 656 & 131 & -14 & -2513 & 188 & 16 & -206 \\
-52 & -1 & 1 & 52 & 0 & 0 & 0 & 0 & 0 & 0 & -3672 & -16284 & 3536 & 2431 & -376 & -31 & 12512 & 612 & -32 & -766 \\
3 & 0 & 0 & -3 & 0 & 0 & 0 & 0 & 0 & 0 & -27 & -38 & 39 & -4 & -11 & 6 & 13 & -18 & 4 & 12 \\
-18 & 0 & 0 & 18 & 0 & 0 & 0 & 0 & 0 & 0 & -1268 & -201 & 702 & -366 & 54 & 9 & -1242 & 202 & 2 & -396 \\
50 & 0 & 0 & -50 & 0 & 0 & 0 & 0 & 0 & 0 & -24507 & 8010 & 1365 & -2100 & 615 & -42 & 11100 & -210 & -11 & 5180 \\
-10 & 0 & 0 & 10 & 0 & 0 & 0 & 0 & 0 & 0 & -372 & 255 & -130 & 46 & -10 & 1 & 110 & -45 & 10 & -141
\end{pmatrix}.
$$

and characters are

$$\chi_0 = q^{\frac{41}{24}}\left(1 + q^2 + q^3 + 2q^4 + 2q^5 + 4q^6 + \cdots\right),$$

$$\chi_1 = q^{\frac{167}{24}}\left(1 + q + 2q^2 + 3q^3 + 4q^4 + 6q^5 + 9q^6 + \cdots\right),$$

$$\chi_2 = q^{-\frac{1}{24}}\left(1 + q + 2q^2 + 3q^3 + 5q^4 + 7q^5 + 11q^6 + \cdots\right),$$

$$\chi_3 = q^{\frac{41}{24}}\left(1 + 2q + 3q^2 + 5q^3 + 8q^4 + 12q^5 + 18q^6 + \cdots\right),$$

$$\chi_4 = q^{-\frac{1}{168}}\left(1 + q + 2q^2 + 3q^3 + 5q^4 + 7q^5 + 12q^6 + \cdots\right),$$

$$\chi_5 = q^{\frac{17}{168}}\left(1 + q + 2q^2 + 3q^3 + 5q^4 + 8q^5 + 12q^6 + \cdots\right),$$

$$\chi_6 = q^{\frac{47}{168}}\left(1 + q + 2q^2 + 3q^3 + 6q^4 + 8q^5 + 13q^6 + \cdots\right),$$

$$\chi_7 = q^{\frac{89}{168}}\left(1 + q + 2q^2 + 4q^3 + 6q^4 + 9q^5 + 14q^6 + \cdots\right),$$

$$\chi_8 = q^{\frac{143}{168}}\left(1 + q + 3q^2 + 4q^3 + 7q^4 + 10q^5 + 16q^6 + \cdots\right),$$

$$\chi_9 = q^{\frac{209}{168}}\left(1 + 2q + 3q^2 + 5q^3 + 8q^4 + 12q^5 + 18q^6 + \cdots\right),$$

$$\chi_{10} = q^{-\frac{5}{264}}\left(1 + q + 2q^2 + 3q^3 + 5q^4 + 7q^5 + 11q^6 + \cdots\right),$$

$$\chi_{11} = q^{\frac{13}{264}}\left(1 + q + 2q^2 + 3q^3 + 5q^4 + 7q^5 + 11q^6 + \cdots\right),$$

$$\chi_{12} = q^{\frac{43}{264}}\left(1 + q + 2q^2 + 3q^3 + 5q^4 + 7q^5 + 11q^6 + \cdots\right),$$

$$\chi_{13} = q^{\frac{85}{264}}\left(1 + q + 2q^2 + 3q^3 + 5q^4 + 7q^5 + 11q^6 + \cdots\right),$$

$$\chi_{14} = q^{\frac{139}{264}}\left(1 + q + 2q^2 + 3q^3 + 5q^4 + 7q^5 + 12q^6 + \cdots\right),$$

$$\chi_{15} = q^{\frac{205}{264}}\left(1 + q + 2q^2 + 3q^3 + 5q^4 + 8q^5 + 12q^6 + \cdots\right),$$

$$\chi_{16} = q^{\frac{283}{264}}\left(1 + q + 2q^2 + 3q^3 + 6q^4 + 8q^5 + 13q^6 + \cdots\right),$$

$$\chi_{17} = q^{\frac{373}{264}}\left(1 + q + 2q^2 + 4q^3 + 6q^4 + 9q^5 + 14q^6 + \cdots\right),$$

$$\chi_{18} = q^{\frac{475}{264}}\left(1 + q + 3q^2 + 4q^3 + 7q^4 + 10q^5 + 16q^6 + \cdots\right),$$

$$\chi_{19} = q^{\frac{589}{264}}\left(1 + 2q + 3q^2 + 5q^3 + 8q^4 + 12q^5 + 18q^6 + \cdots\right).$$

$$(97)$$

**The Haagerup RCFT $\mathcal{R}_{k=10}$**   For $k = 10$, $\Lambda$ and $\chi$ become

$$\Lambda = \mathrm{diag}\left\{-\tfrac{1}{24}, -\tfrac{1}{24}, -\tfrac{1}{24}, -\tfrac{1}{24}, -\tfrac{97}{96}, -\tfrac{11}{12}, -\tfrac{73}{96}, -\tfrac{13}{24}, -\tfrac{25}{96}, -\tfrac{11}{12}, -\tfrac{49}{96}, -\tfrac{49}{48}, -\tfrac{23}{24}, -\tfrac{41}{48}, -\tfrac{17}{24}, -\tfrac{25}{48}, -\tfrac{7}{24}, -\tfrac{1}{48}, -\tfrac{17}{24}, -\tfrac{17}{48}, -\tfrac{23}{24}, -\tfrac{25}{48}\right\},$$

$$\chi = \begin{pmatrix}
1 & 0 & 0 & 0 & -41600 & 22464 & -7488 & 1300 & -64 & 22592 & -960 & 36288 & -24300 & 12000 & -4104 & 864 & -84 & 0 & -4158 & 160 & -24300 & 864 \\
0 & 1 & 0 & 0 & 41600 & 22592 & 7488 & 1300 & 64 & 22464 & 960 & -36288 & -24300 & -12000 & -4158 & -864 & -84 & 0 & -4104 & -160 & -24300 & -864 \\
0 & 0 & 1 & 0 & 41600 & 22592 & 7488 & 1300 & 64 & 22464 & 960 & 36288 & 24300 & 12000 & 4158 & 864 & 84 & 0 & 4104 & 160 & 24300 & 864 \\
0 & 0 & 0 & 1 & -41600 & 22464 & -7488 & 1300 & -64 & 22592 & -960 & -36288 & 24300 & -12000 & 4104 & -864 & 84 & 0 & 4158 & -160 & 24300 & -864 \\
-1 & 1 & 1 & -1 & -2 & 16 & -14 & 0 & -7 & -16 & -17 & 0 & 0 & 0 & 0 & 0 & 0 & 0 & 0 & 0 & 0 & 0 \\
0 & 1 & 1 & 0 & 48 & -7 & -40 & -26 & -8 & -15 & 8 & 0 & 0 & 0 & 0 & 0 & 0 & 0 & 0 & 0 & 0 & 0 \\
-1 & 1 & 1 & -1 & 147 & -240 & -182 & 0 & 12 & 240 & -35 & 0 & 0 & 0 & 0 & 0 & 0 & 0 & 0 & 0 & 0 & 0 \\
1 & 1 & 1 & 1 & 0 & -2048 & 0 & 273 & 0 & -2048 & 0 & 0 & 0 & 0 & 0 & 0 & 0 & 0 & 0 & 0 & 0 & 0 \\
-1 & 1 & 1 & -1 & -8625 & -9200 & 4575 & 0 & -69 & 9200 & 322 & 0 & 0 & 0 & 0 & 0 & 0 & 0 & 0 & 0 & 0 & 0 \\
1 & 0 & 0 & 1 & -48 & -15 & 40 & -26 & 8 & -7 & -8 & 0 & 0 & 0 & 0 & 0 & 0 & 0 & 0 & 0 & 0 & 0 \\
0 & 0 & 0 & 0 & -3675 & 1792 & -441 & 0 & 21 & -1792 & 256 & 0 & 0 & 0 & 0 & 0 & 0 & 0 & 0 & 0 & 0 & 0 \\
1 & -1 & 1 & -1 & 0 & 0 & 0 & 0 & 0 & 0 & 0 & -7 & 0 & -2 & 10 & -13 & 0 & -1 & -10 & -10 & 0 & -13 \\
0 & -1 & 1 & 0 & 0 & 0 & 0 & 0 & 0 & 0 & 0 & 21 & 0 & -1 & -12 & -19 & -7 & -1 & -22 & -5 & 23 & 6 \\
1 & -1 & 1 & -1 & 0 & 0 & 0 & 0 & 0 & 0 & 0 & 81 & 0 & -77 & -54 & -27 & 0 & 0 & 54 & 3 & 0 & -27 \\
0 & -1 & 1 & 0 & 0 & 0 & 0 & 0 & 0 & 0 & 0 & 320 & -276 & -352 & -87 & 32 & 20 & 0 & -100 & -32 & -276 & 32 \\
1 & -1 & 1 & -1 & 0 & 0 & 0 & 0 & 0 & 0 & 0 & 770 & -2048 & -1001 & 330 & 209 & 0 & -1 & -330 & 44 & 2048 & -66 \\
-1 & -1 & 1 & 1 & 0 & 0 & 0 & 0 & 0 & 0 & 0 & 0 & -11178 & 0 & 2430 & 0 & -77 & 0 & 2430 & 0 & -11178 & 0 \\
1 & -1 & 1 & -1 & 0 & 0 & 0 & 0 & 0 & 0 & 0 & -21252 & -47104 & 18952 & 4554 & -1819 & 0 & 0 & -4554 & -253 & 47104 & 506 \\
-1 & 0 & 0 & 1 & 0 & 0 & 0 & 0 & 0 & 0 & 0 & -320 & -276 & 352 & -100 & -32 & 20 & 0 & -87 & 32 & -276 & -32 \\
0 & 0 & 0 & 0 & 0 & 0 & 0 & 0 & 0 & 0 & 0 & -7371 & 0 & 2925 & -1728 & 351 & 0 & 0 & 1728 & -74 & 0 & 351 \\
-1 & 0 & 0 & 1 & 0 & 0 & 0 & 0 & 0 & 0 & 0 & -21 & 23 & 1 & -22 & 19 & -7 & 1 & -12 & 5 & 0 & -6 \\
0 & 0 & 0 & 0 & 0 & 0 & 0 & 0 & 0 & 0 & 0 & -3045 & 2048 & -999 & 320 & -53 & 0 & 1 & -320 & 54 & -2048 & 222
\end{pmatrix},$$

and the characters are

$$\chi_0 = q^{\frac{47}{24}}\left(1 + q^2 + q^3 + 2q^4 + 2q^5 + 4q^6 + \cdots\right),$$

$$\chi_1 = q^{\frac{191}{24}}\left(1 + q + 2q^2 + 3q^3 + 4q^4 + 6q^5 + 9q^6 + \cdots\right),$$

$$\chi_2 = q^{-\frac{1}{24}}\left(1 + q + 2q^2 + 3q^3 + 5q^4 + 7q^5 + 11q^6 + \cdots\right),$$

$$\chi_3 = q^{\frac{47}{24}}\left(1 + 2q + 3q^2 + 5q^3 + 8q^4 + 12q^5 + 18q^6 + \cdots\right),$$

$$\chi_4 = q^{-\frac{1}{96}}\left(1 + q + 2q^2 + 3q^3 + 5q^4 + 7q^5 + 11q^6 + \cdots\right),$$

$$\chi_5 = q^{\frac{1}{12}}\left(1 + q + 2q^2 + 3q^3 + 5q^4 + 7q^5 + 12q^6 + \cdots\right),$$

$$\chi_6 = q^{\frac{23}{96}}\left(1 + q + 2q^2 + 3q^3 + 5q^4 + 8q^5 + 12q^6 + \cdots\right),$$

$$\chi_7 = q^{\frac{11}{24}}\left(1 + q + 2q^2 + 3q^3 + 6q^4 + 8q^5 + 13q^6 + \cdots\right),$$

$$\chi_8 = q^{\frac{71}{96}}\left(1 + q + 2q^2 + 4q^3 + 6q^4 + 9q^5 + 14q^6 + \cdots\right),$$

$$\chi_9 = q^{\frac{13}{12}}\left(1 + q + 3q^2 + 4q^3 + 7q^4 + 10q^5 + 16q^6 + \cdots\right),$$

$$\chi_{10} = q^{\frac{143}{96}}\left(1 + 2q + 3q^2 + 5q^3 + 8q^4 + 12q^5 + 18q^6 + \cdots\right),$$

$$\chi_{11} = q^{-\frac{1}{48}}\left(1 + q + 2q^2 + 3q^3 + 5q^4 + 7q^5 + 11q^6 + \cdots\right),$$

$$\chi_{12} = q^{\frac{1}{24}}\left(1 + q + 2q^2 + 3q^3 + 5q^4 + 7q^5 + 11q^6 + \cdots\right),$$

$$\chi_{13} = q^{\frac{7}{48}}\left(1 + q + 2q^2 + 3q^3 + 5q^4 + 7q^5 + 11q^6 + \cdots\right),$$

$$\chi_{14} = q^{\frac{7}{24}}\left(1 + q + 2q^2 + 3q^3 + 5q^4 + 7q^5 + 11q^6 + \cdots\right),$$

$$\chi_{15} = q^{\frac{23}{48}}\left(1 + q + 2q^2 + 3q^3 + 5q^4 + 7q^5 + 11q^6 + \cdots\right),$$

$$\chi_{16} = q^{\frac{17}{24}}\left(1 + q + 2q^2 + 3q^3 + 5q^4 + 7q^5 + 12q^6 + \cdots\right),$$

$$\chi_{17} = q^{\frac{47}{48}}\left(1 + q + 2q^2 + 3q^3 + 5q^4 + 8q^5 + 12q^6 + \cdots\right),$$

$$\chi_{18} = q^{\frac{31}{24}}\left(1 + q + 2q^2 + 3q^3 + 6q^4 + 8q^5 + 13q^6 + \cdots\right),$$

$$\chi_{19} = q^{\frac{79}{48}}\left(1 + q + 2q^2 + 4q^3 + 6q^4 + 9q^5 + 14q^6 + \cdots\right),$$

$$\chi_{20} = q^{\frac{49}{24}}\left(1 + q + 3q^2 + 4q^3 + 7q^4 + 10q^5 + 16q^6 + \cdots\right),$$

$$\chi_{21} = q^{\frac{119}{48}}\left(1 + 2q + 3q^2 + 5q^3 + 8q^4 + 12q^5 + 18q^6 + \cdots\right).$$

(98)

**The Haagerup RCFT $\mathcal{R}_{k=11}$**  Finally, the two constant matrices for $k = 11$ are

$$\Lambda = \mathrm{diag}\left\{-\tfrac{19}{24}, -\tfrac{1}{24}, -\tfrac{1}{24}, -\tfrac{19}{24}, -\tfrac{73}{72}, -\tfrac{67}{72}, -\tfrac{19}{24}, -\tfrac{43}{72}, -\tfrac{25}{72}, -\tfrac{1}{24}, -\tfrac{49}{72}, -\tfrac{19}{72}, -\tfrac{319}{312}, -\tfrac{301}{312}, -\tfrac{271}{312}, -\tfrac{229}{312}, -\tfrac{175}{312}, -\tfrac{109}{312}, -\tfrac{31}{312}, -\tfrac{253}{312}, -\tfrac{151}{312}, -\tfrac{37}{312}, -\tfrac{223}{312}, \tfrac{85}{312}\right\},$$

$$\chi = \begin{pmatrix}
-70 & 0 & 0 & -7 & -162 & 135 & -85 & 36 & -9 & 1 & -45 & 9 & 141 & -112 & 86 & -58 & 28 & -8 & 1 & -64 & 16 & -2 & 44 & -8 \\
8132 & 1 & 0 & 772 & 40176 & 23301 & 8924 & 1953 & 180 & 1 & 3816 & 63 & -35300 & -24304 & -12772 & -4825 & -1204 & -150 & -4 & -8480 & -584 & -5 & -4180 & -59 \\
772 & 0 & 1 & 8132 & 40176 & 23301 & 8924 & 1953 & 180 & 1 & 3816 & 63 & 35300 & 24304 & 12772 & 4825 & 1204 & 150 & 4 & 8480 & 584 & 5 & 4180 & 59 \\
-7 & 0 & 0 & -70 & -162 & 135 & -85 & 36 & -9 & 1 & -45 & 9 & -141 & 112 & -86 & 58 & -28 & 8 & -1 & 64 & -16 & 2 & -44 & 8 \\
-7 & 1 & 1 & -7 & 7 & -5 & 7 & -8 & -5 & -1 & -20 & -7 & 0 & 0 & 0 & 0 & 0 & 0 & 0 & 0 & 0 & 0 & 0 & 0 \\
20 & 1 & 1 & 20 & 32 & -2 & -20 & -29 & -12 & -1 & -8 & 1 & 0 & 0 & 0 & 0 & 0 & 0 & 0 & 0 & 0 & 0 & 0 & 0 \\
-85 & 1 & 1 & -85 & 162 & -135 & -162 & -36 & 9 & 1 & 45 & -9 & 0 & 0 & 0 & 0 & 0 & 0 & 0 & 0 & 0 & 0 & 0 & 0 \\
344 & 1 & 1 & 344 & 304 & -1279 & -344 & 195 & 40 & -1 & -384 & 2 & 0 & 0 & 0 & 0 & 0 & 0 & 0 & 0 & 0 & 0 & 0 & 0 \\
-1735 & 1 & 1 & -1735 & -2138 & -7285 & 1735 & 764 & -100 & -1 & 1618 & 13 & 0 & 0 & 0 & 0 & 0 & 0 & 0 & 0 & 0 & 0 & 0 & 0 \\
8924 & 1 & 1 & 8924 & -23301 & 17828 & -1953 & -180 & 2 & -3816 & -63 & 0 & 0 & 0 & 0 & 0 & 0 & 0 & 0 & 0 & 0 & 0 & 0 & 0 \\
-189 & 0 & 0 & -189 & -735 & 112 & 189 & -140 & 35 & 0 & -31 & 20 & 0 & 0 & 0 & 0 & 0 & 0 & 0 & 0 & 0 & 0 & 0 & 0 \\
2808 & 0 & 0 & 2808 & -19760 & 9984 & -2808 & 260 & 40 & 0 & 2080 & -61 & 0 & 0 & 0 & 0 & 0 & 0 & 0 & 0 & 0 & 0 & 0 & 0 \\
9 & -1 & 1 & -9 & 0 & 0 & 0 & 0 & 0 & 0 & 0 & 6 & -2 & 5 & -4 & -1 & -4 & -1 & -6 & -11 & -2 & -11 & -6 \\
-4 & -1 & 1 & 4 & 0 & 0 & 0 & 0 & 0 & 0 & 0 & 18 & 7 & -2 & -11 & -14 & -10 & -2 & -20 & -12 & -1 & 10 & 1 \\
35 & -1 & 1 & -35 & 0 & 0 & 0 & 0 & 0 & 0 & 0 & 84 & 0 & -47 & -56 & -28 & -6 & 0 & 48 & 15 & 0 & -12 & -8 \\
-108 & -1 & 1 & 108 & 0 & 0 & 0 & 0 & 0 & 0 & 0 & 262 & -147 & -278 & -133 & 14 & 19 & 2 & 24 & -44 & -4 & -66 & 4 \\
386 & -1 & 1 & -386 & 0 & 0 & 0 & 0 & 0 & 0 & 0 & 790 & -1316 & -982 & 80 & 203 & 30 & -2 & -812 & -17 & 4 & 374 & -12 \\
-1408 & -1 & 1 & 1408 & 0 & 0 & 0 & 0 & 0 & 0 & 0 & 1212 & -7472 & -1572 & 1769 & 364 & -83 & -4 & 1824 & 312 & -3 & -1716 & 4 \\
5534 & -1 & 1 & -5534 & 0 & 0 & 0 & 0 & 0 & 0 & 0 & -6096 & -34104 & 6652 & 6508 & -1064 & -190 & 5 & 7240 & -848 & -4 & 6160 & 16 \\
-52 & 0 & 0 & 52 & 0 & 0 & 0 & 0 & 0 & 0 & 0 & -74 & -140 & 114 & 12 & -42 & 12 & 2 & -113 & 12 & 3 & -66 & -2 \\
624 & 0 & 0 & -624 & 0 & 0 & 0 & 0 & 0 & 0 & 0 & -2304 & -1104 & 1832 & -684 & -36 & 64 & -3 & 192 & 145 & -4 & 297 & 24 \\
-5044 & 0 & 0 & 5044 & 0 & 0 & 0 & 0 & 0 & 0 & 0 & -33638 & 3794 & 9558 & -6456 & 1554 & -81 & -2 & 10152 & -852 & 7 & 738 & -68 \\
117 & 0 & 0 & -117 & 0 & 0 & 0 & 0 & 0 & 0 & 0 & -531 & 294 & -47 & -72 & 63 & -22 & 3 & -206 & 27 & 2 & -98 & 18 \\
-2236 & 0 & 0 & 2236 & 0 & 0 & 0 & 0 & 0 & 0 & 0 & -16890 & 11116 & -5318 & 1720 & -322 & 20 & 2 & -1766 & 292 & -4 & 2398 & -57
\end{pmatrix},$$

and the characters are

$$\chi_0 = q^{\frac{53}{24}} \left(1 + q^2 + q^3 + 2q^4 + 2q^5 + 4q^6 + \cdots\right),$$

$$\chi_1 = q^{\frac{215}{24}} \left(1 + q + 2q^2 + 3q^3 + 4q^4 + 6q^5 + 9q^6 + \cdots\right),$$

$$\chi_2 = q^{-\frac{1}{24}} \left(1 + q + 2q^2 + 3q^3 + 5q^4 + 7q^5 + 11q^6 + \cdots\right),$$

$$\chi_3 = q^{\frac{53}{24}} \left(1 + 2q + 3q^2 + 5q^3 + 8q^4 + 12q^5 + 18q^6 + \cdots\right),$$

$$\chi_4 = q^{-\frac{1}{72}} \left(1 + q + 2q^2 + 3q^3 + 5q^4 + 7q^5 + 11q^6 + \cdots\right),$$

$$\chi_5 = q^{\frac{5}{72}} \left(1 + q + 2q^2 + 3q^3 + 5q^4 + 7q^5 + 11q^6 + \cdots\right),$$

$$\chi_6 = q^{\frac{5}{24}} \left(1 + q + 2q^2 + 3q^3 + 5q^4 + 7q^5 + 12q^6 + \cdots\right),$$

$$\chi_7 = q^{\frac{29}{72}} \left(1 + q + 2q^2 + 3q^3 + 5q^4 + 8q^5 + 12q^6 + \cdots\right),$$

$$\chi_8 = q^{\frac{47}{72}} \left(1 + q + 2q^2 + 3q^3 + 6q^4 + 8q^5 + 13q^6 + \cdots\right),$$

$$\chi_9 = q^{\frac{23}{24}} \left(1 + q + 2q^2 + 4q^3 + 6q^4 + 9q^5 + 14q^6 + \cdots\right),$$

$$\chi_{10} = q^{\frac{95}{72}} \left(1 + q + 3q^2 + 4q^3 + 7q^4 + 10q^5 + 16q^6 + \cdots\right),$$

$$\chi_{11} = q^{\frac{125}{72}} \left(1 + 2q + 3q^2 + 5q^3 + 8q^4 + 12q^5 + 18q^6 + \cdots\right),$$

$$\chi_{12} = q^{-\frac{7}{312}} \left(1 + q + 2q^2 + 3q^3 + 5q^4 + 7q^5 + 11q^6 + \cdots\right),$$

$$\chi_{13} = q^{\frac{11}{312}} \left(1 + q + 2q^2 + 3q^3 + 5q^4 + 7q^5 + 11q^6 + \cdots\right),$$

$$\chi_{14} = q^{\frac{41}{312}} \left(1 + q + 2q^2 + 3q^3 + 5q^4 + 7q^5 + 11q^6 + \cdots\right),$$

$$\chi_{15} = q^{\frac{83}{312}} \left(1 + q + 2q^2 + 3q^3 + 5q^4 + 7q^5 + 11q^6 + \cdots\right),$$

$$\chi_{16} = q^{\frac{137}{312}} \left(1 + q + 2q^2 + 3q^3 + 5q^4 + 7q^5 + 11q^6 + \cdots\right),$$

$$\chi_{17} = q^{\frac{203}{312}} \left(1 + q + 2q^2 + 3q^3 + 5q^4 + 7q^5 + 11q^6 + \cdots\right),$$

$$\chi_{18} = q^{\frac{281}{312}} \left(1 + q + 2q^2 + 3q^3 + 5q^4 + 7q^5 + 12q^6 + \cdots\right),$$

$$\chi_{19} = q^{\frac{371}{312}} \left(1 + q + 2q^2 + 3q^3 + 5q^4 + 8q^5 + 12q^6 + \cdots\right),$$

$$\chi_{20} = q^{\frac{473}{312}} \left(1 + q + 2q^2 + 3q^3 + 6q^4 + 8q^5 + 13q^6 + \cdots\right),$$

$$\chi_{21} = q^{\frac{587}{312}} \left(1 + q + 2q^2 + 4q^3 + 6q^4 + 9q^5 + 14q^6 + \cdots\right),$$

$$\chi_{22} = q^{\frac{713}{312}} \left(1 + q + 3q^2 + 4q^3 + 7q^4 + 10q^5 + 16q^6 + \cdots\right),$$

$$\chi_{23} = q^{\frac{851}{312}} \left(1 + 2q + 3q^2 + 5q^3 + 8q^4 + 12q^5 + 18q^6 + \cdots\right).$$

## 4 Conclusion and future works

We have explored the non-unitary version of bulk-boundary correspondence with concrete examples. For those examples, we have confirmed that some basic dictionaries work well even in the non-unitary cases. By applying the correspondence, we obtain a new class of non-unitary RCFTs from known bulk rank-0 theories. By applying the correspondence in the reverse direction, one may construct new classes of 3D rank-0 SCFTs from known non-unitary RCFTs as studied in [36].

Here we list a set of related problems which we hope to address in the near future.

**Chiral algebra of $\mathcal{R}_k$** In this paper, we analyze the correspondence using modular data and characters. Another important mathematical object of a 2D RCFT is its underlying chiral

algebra. Each character is associated to an irreducible representation of the algebra. It would be interesting to identify the chiral algebra of the $\mathcal{R}_k$ by analyzing the boundary operator algebra of the twisted S-fold SCFT using the techniques developed in [9, 33, 45, 68–75]. For $k = 3$, the $\mathcal{R}_k$ is a product of two Virasoro minimal models, $M(3, 5)$ and $M(2, 5)$, and the chiral algebra is expected to be two copies of Virasoro algebra.

**Hecke transformation to unitary RCFTs**   Hecke transformation acts on a vvmf to another vvmf. Applying the transformation, we may hope to find other admissible RCFT characters from the characters of $\mathcal{R}_k$ we obtained. For some cases, the Hecke transform relates non-unitary RCFTs to unitary ones. It would be interesting to see if one can generate a new class of admissible characters of unitary RCFTs by applying Heck transformations to the characters of $\mathcal{R}_k$.

**More general classes of Haagerup RCFTs**   When $k = 4m^2 + 4m + 3$ ($m \in \mathbb{Z}_{\geq 0}$), the modular data of $\mathcal{S}_k$ is related to an unitary Haagerup-Izumi modular data $\mathcal{D}^\omega \mathrm{Hg}_{2m+1}$ with $\omega = \pm 2$ by a Hecke transformation (modulo decoupled $U(1)_{\pm 2}$). One may also consider non-unitary modular data obtained by acting a Hecke transformation on $\mathcal{D}^\omega \mathrm{Hg}_{2m+1}$ with $\omega \neq \pm 2$. It would be interesting to find 3D rank-0 SCFTs which realize non-unitary TQFTs with the modular data via a topological twisting. Since the existence of non-unitary 3D TQFTs with the modular data is obscure, there is no guarantee that such rank-0 SCFTs should exit.

# Acknowledgments

We would like to thank Heeyeon Kim, Kimyeong Lee, and Brandon Rayhaun for the useful discussion.

**Funding information**   The work of DG and DK is supported in part by the National Research Foundation of Korea grant NRF-2022R1C1C1011979. DG also acknowledges support by Creative-Pioneering Researchers Program through Seoul National University. SL is supported in part by KIAS Grant PG056502.

# A   Dual Abelian description of $\mathcal{S}_k$ from 3D-3D correspondence

3D-3D correspondence predicts following IR duality [6, 18, 20]

$$\left( \widetilde{\mathcal{S}}_k \text{ in (1) and (26)} \right) \simeq \mathcal{T}_{DGG}[(\Sigma_{1,1} \times S^1)_{\varphi = ST^k \simeq LR^{k-2}}; A = \lambda]. \tag{A.1}$$

Here $\mathcal{T}_{DGG}[N; A]$ is the 3D $\mathcal{N} = 2$ gauge theory labeled by an ideally triangulated 3-manifold $N$ with a torus boundary proposed in [19]. The theory also depends on a choice of primitive boundary 1-cycle $A \in H_1(\partial N, \mathbb{Z})$. Here $(\Sigma_{1,1} \times S^1)_\varphi$ denotes the once-punctured torus bundle with a monodromy $\varphi \in SL(2, \mathbb{Z})$ (mapping class group of $\Sigma_{1,1}$, once-punctured torus):

$$(\Sigma_{1,1} \times S^1)_\varphi = (\Sigma_{1,1} \times [0, 1]) / \sim,$$
$$\text{with the equivalence relation} \quad (x \in \Sigma_{1,1}, 0) \sim (\varphi(x), 1). \tag{A.2}$$

$A = \lambda$ is a boundary 1-cycle circling the puncture on $\Sigma_{1,1}$. The $SL(2, \mathbb{Z})$ matrices $S, T = R$ and $L$ are

$$S = \begin{pmatrix} 0 & 1 \\ -1 & 0 \end{pmatrix}, \qquad T = R = \begin{pmatrix} 1 & 0 \\ 1 & 1 \end{pmatrix}, \qquad L = \begin{pmatrix} 1 & 1 \\ 0 & 1 \end{pmatrix}. \tag{A.3}$$

Topology of the torus bundle depends only on the conjugacy class of $\varphi$ and note that $ST^k \simeq LR^{k-2}$ up to a conjugation.

One convenient way to identify a field theory description of $\mathcal{T}_{DGG}[N;A]$ is calculating $SL(2,\mathbb{C})$ Chern-Simons theory partition functions $\mathcal{Z}^{SL(2,\mathbb{C}) \text{ on } (N,A)}_{(k,\sigma)}(M_A)$ on the 3-manifold $N$ with various $k \in \mathbb{Z}$:

$$\mathcal{Z}^{SL(2,\mathbb{C}) \text{ on } (N,A)}_{(k,\sigma)}(M_A)$$
$$= \int \frac{[D\mathcal{A}][D\bar{\mathcal{A}}]}{(\text{gauge})}\Bigg|_{b.c} \exp\left[i\left(\int_N \frac{k+\sigma}{8\pi}\text{Tr}\left(\mathcal{A} \wedge d\mathcal{A} + \frac{2}{3}\mathcal{A}^3\right) + \frac{k-\sigma}{8\pi}\text{Tr}\left(\bar{\mathcal{A}} \wedge d\bar{\mathcal{A}} + \frac{2}{3}\bar{\mathcal{A}}^3\right)\right)\right],$$

with the boundary condition : $P\exp\left(\oint_A \mathcal{A}\right)\Bigg|_{\partial N} \sim \begin{pmatrix} e^{M_A/2} & * \\ 0 & e^{-M_A/2} \end{pmatrix} \in SL(2,\mathbb{C}).$ (A.4)

According to 3D-3D relations [18, 19, 76–80], the CS partition functions with $k(\in \mathbb{Z})$ and $\sigma(\in \mathbb{R} \text{ or } i\mathbb{R})$ compute various SUSY partition functions of $\mathcal{T}_{DGG}[N;A]$. Especially when with $k = 1$ and $\sigma = \frac{1-b^2}{1+b^2}$, the CS partition function is related to the squashed 3-sphere partition function of $\mathcal{T}_{DGG}[N;A]$ in the following way

$$\left(\mathcal{Z}_{S^3_b}(m,\nu) \text{ of } \mathcal{T}_{DGG}[N;A]\right) = \sqrt{\frac{|H_1(N,\mathbb{Z}_2)|}{2}} \mathcal{Z}^{SL(2,\mathbb{C}) \text{ on } (N,A)}_{(1,\frac{1-b^2}{1+b^2})}(M_A), \quad \text{with}$$
$$m + \left(i\pi + \frac{\hbar}{2}\right)\nu = \begin{cases} \frac{M_A}{2}, & A \in \text{Ker}\left(i_* : H_1(\partial N,\mathbb{Z}) \to H_1(N,\mathbb{Z}_2)\right), \\ M_A, & \text{otherwise.} \end{cases}$$ (A.5)

The $\mathcal{T}_{DGG}[N;A]$ has a $U(1)_A$ symmetry associated to the torus boundary and $(m,\nu)$ is the (rescaled real mass, R-symmetry mixing parameter) of the flavor symmetry. The boundary 1-cycle $\lambda$ of $N = (\Sigma_{1,1} \times S^1)_\varphi$ belongs to the kernel of the map $i_* : H_1(\partial N,\mathbb{Z}) \to H_1(N,\mathbb{Z}_2)$. As we will see below, the Chern-Simons partition function can be given as a finite dimensional integral which is identical to a squashed 3-sphere partition function localization integral of a 3D $\mathcal{N} = 2$ gauge theory, which can be identified as $\mathcal{T}_{DGG}[N,A]$.

The Chern-Simons partition function can be computed using the so-called state-integral model based on an ideal triangulation of $N$ [80–86]. There is a canonical way to ideally triangulate the once-puncture torus bundle $(\Sigma_{1,1} \times S^1)_{\varphi=LR^{n=k-2}}$ with $(n+1)$ ideal tetrahedra [87]. Farey tessellation from the triangulation is given as figure 1. From the farey tessellation, we can read off the following gluing data

$$M_\mu = Y'' - \sum_{i=1}^n X_i,$$
$$\frac{M_\lambda}{2} = X'_n + X''_{n-1} - X_n - Y = \left(i\pi + \frac{\hbar}{2}\right) - 2X_n - X''_n + X''_{n-1} - Y,$$
$$C_1 = 2Y'' + 2X'_n + X''_1 + X''_{n-1} - 2\left(i\pi + \frac{\hbar}{2}\right) = 2Y'' - 2X_n + X''_1 + X''_{n-1} - 2X''_n,$$
$$C_2 = Y + 2X'_1 + X''_2 - 2\left(i\pi + \frac{\hbar}{2}\right) = Y - 2X_1 - 2X''_1 + X''_2,$$
$$\vdots$$
$$C_{n-1} = X''_{n-3} + 2X'_{n-2} + X''_{n-1} - 2\left(i\pi + \frac{\hbar}{2}\right) = -2X_{n-2} + X''_{n-3} - 2X''_{n-2} + X''_{n-1},$$
$$C_n = X''_{n-2} + 2X'_{n-1} + X''_n - 2\left(i\pi + \frac{\hbar}{2}\right) = -2X_{n-1} + X''_{n-2} - 2X''_{n-1} + X''_n,$$
$$C_{n+1} = 2Y' + Y + X''_n + 2\sum_{i=1}^n X_i - 2\left(i\pi + \frac{\hbar}{2}\right) = -Y - 2Y'' + X''_n + 2\sum_{i=1}^n X_i.$$ (A.6)

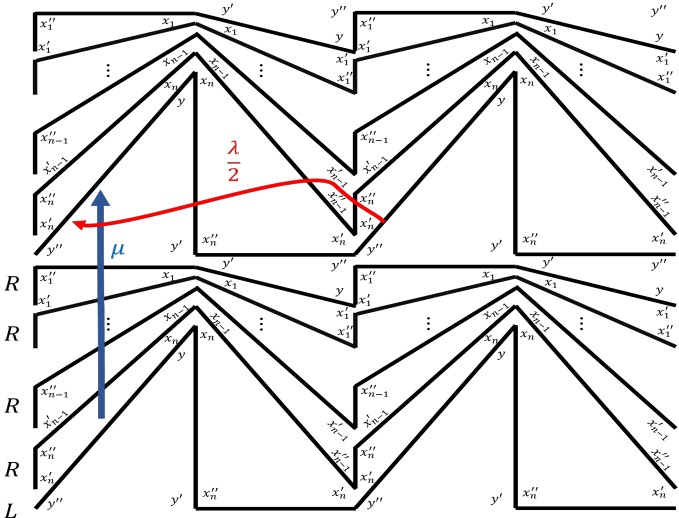

Figure 1: Farey tessellation of once punctured torus bundle with monodromy $LR^n$.

In the construction of $\mathcal{T}_{DGG}$ [19], the 'easy' internal edges, $C_1, \ldots, C_n$, correspond to chiral primary operators appearing in the superpotential (5). Generally, the gluing data is known to have a symplectic structure [88] and one consider following affine symplectic transformation

$$
\begin{pmatrix} \frac{M_\lambda}{2} \\ C_1 \\ \vdots \\ C_n \\ -M_\mu \\ \Gamma_1 \\ \vdots \\ \Gamma_n \end{pmatrix} = \left( \begin{array}{c|c} A & B \\ \hline C & D \end{array} \right) \begin{pmatrix} Y \\ X_1 \\ \vdots \\ X_n \\ Y'' \\ X_1'' \\ \vdots \\ X_n'' \end{pmatrix} - (i\pi + \hbar/2) \begin{pmatrix} -1 \\ 0 \\ \vdots \\ 0 \\ 0 \\ 0 \\ \vdots \\ 0 \end{pmatrix}, \tag{A.7}
$$

where two block matrices $A, B$ are

$$
A_{(n+1)\times(n+1)} = \begin{pmatrix} -1 & 0 & 0 & \cdots & 0 & -2 \\ 0 & 0 & 0 & \cdots & 0 & -2 \\ 1 & -2 & 0 & \cdots & 0 & 0 \\ 0 & 0 & -2 & 0 & \cdots & 0 \\ & & & \vdots & & \\ 0 & 0 & \cdots & 0 & -2 & 0 \end{pmatrix},
$$

$$
B_{(n+1)\times(n+1)} = \begin{pmatrix} 0 & 0 & \cdots & 0 & 0 & 1 & -1 \\ 2 & 1 & 0 & \cdots & 0 & 1 & -2 \\ 0 & -2 & 1 & 0 & 0 & \cdots & 0 \\ 0 & 1 & -2 & 1 & 0 & \cdots & 0 \\ & & & \vdots & & & \\ 0 & \cdots & 0 & 1 & -2 & 1 & 0 \\ 0 & \cdots & 0 & 0 & 1 & -2 & 1 \end{pmatrix},
\tag{A.8}
$$

for $n \geq 3$.[6] Using the symplectic gluing data, the state integral can be given as [85]

$$
\begin{aligned}
&\mathcal{Z}^{SL(2,\mathbb{C}) \text{ on } (N,A)}_{(1,\frac{1-b^2}{1+b^2})}(M_\lambda) \\
&= \frac{1}{\sqrt{\det B}} \int \frac{d^{n+1}\mathbf{Z}}{(2\pi\hbar)^{\frac{n+1}{2}}} \exp\left[\frac{1}{2\hbar}\left(\mathbf{X}^T DB^{-1}\mathbf{X} - 2\mathbf{Z}^T B^{-1}\mathbf{X} + \mathbf{Z}^T B^{-1}A\mathbf{Z}\right)\right]\prod_{i=1}^{n+1}\psi_\hbar(Z_i) \\
&= \frac{e^{f(M_\lambda,\hbar)}}{\sqrt{2}} \int \frac{d^{n+1}\mathbf{Z}}{(2\pi\hbar)^{\frac{n+1}{2}}} \exp\left[\frac{1}{2\hbar}\mathbf{Z}^T B^{-1}A\mathbf{Z} + \frac{1}{\hbar}\left(X - i\pi - \frac{\hbar}{2}\right)\left(\frac{n}{2}Z_1 + \sum_{k=2}^n kZ_{k+1}\right)\right]\prod_{i=1}^{n+1}\psi_\hbar(Z_i).
\end{aligned}
\tag{A.9}
$$

Here $e^{f(M_\lambda,\hbar)}$ is a factor independent of $\mathbf{Z}$ which can be interpreted as a contribution from background mixed Chern-Simons levels of $U(1)_A \times U(1)_R$ symmetries in the field theory $\mathcal{T}_{DGG}[N;A]$. In the last line, we let

$$
\mathbf{X} = \left(\frac{M_\lambda}{2} - i\pi - \frac{\hbar}{2}, C_1, \ldots, C_n\right)\Big|_{C_{1\leq i \leq n}=0}.
\tag{A.10}
$$

The matrix $B^{-1}A$ is

$$
B^{-1}A = \begin{pmatrix}
\frac{n}{2} & 0 & 1 & 2 & 3 & \ldots & n-1 \\
0 & 2 & 2 & 2 & 2 & \ldots & 2 \\
1 & 2 & 4 & 4 & 4 & \ldots & 4 \\
2 & 2 & 4 & 6 & 6 & \ldots & 6 \\
3 & 2 & 4 & 6 & 8 & \ldots & 8 \\
\ldots & \ldots & \ldots & \ldots & \ldots & \ldots & \ldots \\
n-1 & 2 & 4 & 6 & 8 & \ldots & 2n
\end{pmatrix}.
\tag{A.11}
$$

Rescaling the $Z_1$ to $2Z_1$, we have (ignoring the factor $e^{f(M_\lambda,\hbar)}$)

$$
\begin{aligned}
&\mathcal{Z}^{SL(2,\mathbb{C}) \text{ on } (N,A)}_{(1,\frac{1-b^2}{1+b^2})}(M_\lambda) \\
&= \sqrt{2} \int \frac{d^{n+1}\mathbf{Z}}{(2\pi\hbar)^{\frac{n+1}{2}}} \exp\left[\frac{1}{2\hbar}\mathbf{Z}^T K\mathbf{Z} + \frac{1}{\hbar}\left(\frac{M_\lambda}{2} - i\pi - \frac{\hbar}{2}\right)\left(nZ_1 + \sum_{k=2}^n kZ_{k+1}\right)\right]\prod_{i=1}^{n+1}\psi_\hbar(Z_i).
\end{aligned}
\tag{A.12}
$$

Here $K$ is the mixed Chern-Simons level in (4). Then combining the 3D-3D relations in (A.1) and (A.5) and the following facts,

$$
\left(\mathcal{Z}_{S_b^3} \text{ of } \mathcal{S}_k^{/\mathbb{Z}_2}\right) = \begin{cases} 2 \times \left(\mathcal{Z}_{S_b^3} \text{ of } \mathcal{S}_k\right), & k \in 2\mathbb{Z}, \\ \sqrt{2} \times \left(\mathcal{Z}_{S_b^3} \text{ of } \mathcal{S}_k\right), & k \in 2\mathbb{Z}+1, \end{cases}
\tag{A.13}
$$

$$
H_1\left((\Sigma_{1,1} \times S^1)_{\varphi=LR^{n=k-2}}, \mathbb{Z}_2\right) = \begin{cases} \mathbb{Z}_2 \times \mathbb{Z}_2, & k \in 2\mathbb{Z}, \\ \mathbb{Z}_2, & k \in 2\mathbb{Z}+1, \end{cases}
\tag{A.14}
$$

we have

$$
\left(\mathcal{Z}_{S_b^3}(m,\nu) \text{ of } \mathcal{S}_k \text{ in } (1)\right) = \left(\mathcal{Z}_{S_b^3}(m,\nu) \text{ in } (10)\right).
\tag{A.15}
$$

It strongly supports the proposed IR duality between (1) and (2).

---

[6]When $n$ is 1 or 2, the exact expression for $A$ and $B$ is different from (A.8). However, the final results agrees with (A.11) and (A.12) for the two cases.

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
