# Peer review of "A non-unitary bulk-boundary correspondence: Non-unitary Haagerup RCFTs from S-fold SCFTs"

_SciPost Physics, doi:SciPost Phys. 17, 064 (2024)_

## Round 1 · Referee Report · Anonymous (Referee 1) · 2024-4-2

## REPORT

The manuscript under review studies a novel class of 2d non-unitary RCFTs constructed from a topological twisting of some 3d $N = 4$ rank-0 SCFTs. It contains many interesting new results and is well presented. In particular, it makes connections to Haagerup TQFTs and Haagerup CFTs which draws lots of attentions in recent years. I **strongly recommend the manuscript to be published on SciPost** after considering the following suggestions.

(1) Nahm sums already have precise definition in mathematics, e.g. by Don Zagier. The general half-indices formula the author present is not really Nahm sum, because of the $(q)_{2m_1}$ in the denominator. To not cause confusion, I would suggest the authors not to call these as Nahm sums, e.g. in the bottom of page 15. Note that the name "generalized Nahm sums" is also taken to describe the situation when the $K$ matrix is not symmetric.

   On the other hand, it looks to me that some characters of the orbifold Haagerup RCFT are indeed Nahm sums after scaling $m_1$. If it's true, I suggest the authors to mention this.

(2) I was under the impression that all non-unitary Haagerup RCFTs constructed in this way have effective central charge $c_{\text{eff}} = 1$. If this is correct, I suggest the authors to add this clear statement into the manuscript.

(3) It would be helpful to readers if the authors can simply write down how many characters there are for each $k$. For example, in pages 8, the authors can say the $S$-matrix is $(2k + 2) \times (2k + 2)$. Same for page 11.

(4) I would like to know if (and how) the authors prove the $S$ and $T$ matrices in page 8 satisfy the relation $S^2 = (ST)^3 = I$ for **arbitrary** $k$.

(5) Below eq (2.32), the author use $\bar{p}$. It would be better to write down the conductor for which the $\bar{p}$ is defined and relation between the conductor and the $n$ in $\text{Hg}_{2n+1}$ in eq (2.32).

(6) The authors discuss the Haagerup RCFT $\mathcal{R}_{k=4}$ in details, which has 10 characters. It looks to me that this theory is exactly the bosonic theory of $N = 1$ supersymmetric minimal model $SM(12, 2)$, which is non-unitary as expected. The effective $SM(12, 2)$ has NS conformal weights $0, \frac{1}{6}, \frac{1}{2}$. I checked the NS characters are related to the $\mathcal{R}_{k=4}$ characters given in eq (3.52) by

$$\chi_0^{\text{NS}} = \chi_2 + \chi_3,$$
$$\chi_{1/6}^{\text{NS}} = \chi_6 + \chi_8,$$
$$\chi_{1/2}^{\text{NS}} = \chi_0 + \chi_1.$$

Similar for the three R characters. The $\chi_9 - \chi_5 = 1$ is the constant $\widetilde{R}$ character. The authors can write about this if they would like to.

(7) For orbifold Haagerup RCFT $\widetilde{\mathcal{R}}_{k=6}$, I tried eqs (3.22) and (3.23) with the $K$ matrix given in eq (3.24) and $m_1 \in \mathbb{N}/2$ and $m_{2,3,4,5} \in \mathbb{N}$. Unfortunately, I cannot recover the authors'

$q$-series. For both eqs (3.22) and (3.23), I get $q^{1/2}$ series. I would like to know whether the formulas (3.22), (3.23) and (3.24) are precisely presented as the authors compute them.

Besides, it seems that (3.22) and (3.23) give the same $q$-series. I would like to know whether this is a coincidence or there is a reason.

(8) For Haagerup RCFT $\mathcal{R}_{k\geq 5}$, the authors present the $\chi$ matrices which I feel are not directly useful to most readers. It would be better if the authors can present the characters they find even in a mathematica attachment, such that readers can easily follow their work and understand these Haagerup RCFTs.

---

## Round 1 · Referee Report · Anonymous (Referee 2) · 2024-4-9

Report

This paper is a very interesting one, where the authors extend their previous results on obtaining 3d non-unitary TQFT from twisting 3d supersymmetric theories by providing their characters by a combination of gauge theory methods and RCFT methods. It easily meets the criteria to be published on SciPost, but a few minor revisions would be welcomed, on the points listed below.

As the referee 1 already wrote a comprehensive report on the side of the non-unitary RCFT, this referee wants to concentrate mostly on the gauge theory part.

  • In (2.1), it would be nice to mention that the authors use $\mathcal{N}=3$ Chern-Simons term rather than $\mathcal{N}=2$ Chern-Simons term.

  • It would be nice to mention, when going from Sec.2.1.1 to 2.1.2, that the construction (2.1) geometrically corresponds to a once-punctured torus bundle. This fact would be obvious to the practitioners like the authors and this referee but would not be at all obvious.

  • Above (2.4), you might want to say that these are $\mathcal{N}=2$ Chern-Simons terms.

  • In Sec.2.2 or in the introduction, you might want to mention that the Haagerup-Izumi TQFTs are rather exotic in the sense they are rare examples of TQFTs not related to doubles of finite groups and/or affine Lie algebras.

  • The claim (2.16), when considered in reverse, means that the parameters $(m,\nu)$ give deformations of the TQFT data. Is there anything the authors could say about these deformations, other than they exist? For example, do $S$ and $T$ as deformed by $(m,\nu)$ still satisfy the $SL(2,\mathbb{Z})$ relations?

  • Around (2.32), it would be nice to explain the notation, the history and the motivations behind these Haagerup-Izumi modular data and the Hecke/Galois transformations. It doesn't have to be long, but the matter-of-factly presentation here could be improved.

  • In (3.11), do the authors really mean (3.13)? Don't we need rational numbers without mod 1 here?

  • It would be nicer to spell out a.k.a. as "also known as", or vvmf as vector valued modular forms, etc.

  • Around (3.28) and elsewhere, the authors say that they study the S-transformation numerically. How do they concretely do that? Do they evaluate the series around $\tau\sim 1$ ? Does the series for $\chi(-1/\tau)$ and $\chi(\tau)$ both converge at that point?

  • In Sec.3.3, they explain the Bantay-Gannon method in the sense of how it should be performed, but they do not provide any explanation why it should work. Again the referee understands that the interested readers should refer to the original paper, but a short explanation here would not be bad.

  • In the table of $\Xi$ and $\chi$ the authors provide, we see a big block of zeros, say in $\mathcal{R}_{k=8,9,10}$. Is there a simple explanation why?

---

## Round 2 · Referee Report · Anonymous (Referee 4) · 2024-6-24

Report

The authors have successfully addressed the concerns and suggestion. These revisions have significantly improved the clarity of the manuscript. I am pleased to recommend the revised manuscript for publication.

Recommendation

Publish (easily meets expectations and criteria for this Journal; among top 50%)

---

## Round 2 · Referee Report · Anonymous (Referee 3) · 2024-6-24

Report

The improvements made by the authors in v2 are satisfactory and the manuscript can now safely be recommended to be published.

Recommendation

Publish (easily meets expectations and criteria for this Journal; among top 50%)

---

## Round 2 · Author Response

We would like to thank the anonymous referees for their insightful comments and suggestions. We have improved the manuscript according to their suggestions.

---

## Round 2 · List of Changes

Here are the changes of manuscript and replies to the referees’ suggestions.

1. We added a comment below equation (3.15) noting that the first two characters can be expressed as Nahm sums after rescaling m1.

2. We mentioned below equation (3.28) that c_{eff} is always 1.

3. We provided the sizes of the modular matrices in equations (2.19), (2.29), and (2.30).

4. The modular matrices in (2.19) were obtained in the reference [7], where the SL(2,Z) relations were checked for various values of k using Mathematica.

5. We added more explanations about the Haagerup-Izumi modular data below equation (2.32).

6. We added a comment below equation (3.54) noting that the characters of R_{k=4}​ are related to the characters of the supersymmetric N=1 minimal model SM(2,12).

7. We corrected a typo in the K matrix in equation (3.24); the first entry should be 8 instead of 4.

8. In section 3.3, we included the explicit q-series of the characters up to some orders.

9. In equations (2.1) and (2.4), we specified whether we are using N=3 or N=2 CS terms in the gauging.

10. At the beginning of section (2.1.3), we added a paragraph stating that the S-fold SCFTs are associated with once-punctured torus bundles in 3D-3D correspondence.

11. The SL(2,Z) relations of modular matrices are only satisfied in the topological twisting limits, i.e. (m,\nu) = (0, \pm 1). Currently, we do not know how to interpret the deformations from the perspective of non-unitary TQFTs or their boundary RCFTs. This would be an interesting direction for future research.

12. From the bulk computation, we can determine the exponents (Delta_alpha) only modulo 1, and we present the values modulo 1 in equation (3.13). Developing a systematic methodology to fully determine the exponents would be an interesting future work.

13. We replaced "a.k.a" with "also known as" and spelled out "vvmf" as "vector-valued modular form" when it first appeared.

13. We numerically checked the S-transformation property of the characters by evaluating them at ττ near ii, where both q=e^{2\pi i \tau} and tilde{q} =exp(2pi i (-1/\tau)) are smaller than 1. We added a related comment below equation (3.27).

14. Below equation (3.38), we added a brief introduction to the Bantay-Gannon method.

---

## Editorial Decision

published